# Accelerated Likelihood Maximization for Diffusion-based Versatile Content Generation

## Abstract

Generating diverse, coherent, and plausible content from partially given inputs remains a significant challenge for pretrained diffusion models. Existing approaches face clear limitations: training-based approaches offer strong task-specific results but require costly data and computation, and they generalize poorly across tasks. Training-free paradigms are more efficient and broadly applicable, but often fail to produce globally consistent results, as they usually enforce constraints only on observed regions. To address these limitations, we introduce Accelerated Likelihood Maximization (ALM), a novel training-free sampling strategy integrated into the reverse process of diffusion models. ALM explicitly optimizes the unobserved regions by jointly maximizing both conditional and joint likelihoods. This ensures that the generated content is not only faithful to the given input but also globally coherent and plausible. We further incorporate an acceleration mechanism to enable efficient computation. Experimental results demonstrate that ALM consistently outperforms state-of-the-art methods in various data domains and tasks, establishing a powerful, training-free paradigm for versatile content generation.

## 1 Introduction

While naïve use of diffusion models (Ho et al., 2020; Song et al., 2021a;b) has primarily focused on generating fixed-size outputs from scratch, many practical tasks demand a more versatile approach—generating content from partially observed or pre-generated inputs. This paradigm, which we call *versatile content generation*, aims to condition the generation process on the available inputs. The core challenge is enabling diffusion models to go beyond simple synthesis and instead infer missing or extended content—like filling in missing regions (inpainting) or extrapolating beyond observed boundaries (outpainting)—that remains globally coherent with the provided context. This includes a wide range of applications across diverse data domains such as image inpainting (Lugmayr et al., 2022; Zhang et al., 2023; Corneanu et al., 2024; Ju et al., 2024; Zhuang et al., 2024; Avrahami et al., 2023; Manukyan et al., 2025), wide image synthesis (Bar-Tal et al., 2023; Kim et al., 2024a; Lee et al., 2025), human motion in-completion (Cohan et al., 2024; Xie et al., 2024) and long video generation (Kim et al., 2024b; Qiu et al., 2024; Wang et al., 2023; Chen et al., 2023).

Despite extensive research, developing a unified approach that generalizes across such diverse content generation tasks remains a major challenge. Existing training-based methods (Ju et al., 2024; Zhuang et al., 2024; Cohan et al., 2024) can achieve strong task-specific performance, but they require substantial computational training and large-scale datasets, limiting practical deployment. Although recent visual foundation models (Rombach et al., 2022; Podell et al., 2024) offer powerful editing capabilities, they also rely on specialized training pipelines, which restrict flexibility when adapting to new tasks or modalities.

In contrast, our goal is to shift this paradigm; Instead of relying on costly retraining, we seek a fully training-free mechanism that can be applied directly to any pretrained generative model. Such a mechanism would immediately enhance widely used models into flexible, high-fidelity content-editing approaches without additional computation. This motivates the need for a general and widely applicable training-free framework for versatile content generation. Existing training-free approaches, such as diffusion synchronization (Bar-Tal et al., 2023; Kim et al., 2024a; Lee et al., 2025), aim to broaden applicability by leveraging pretrained diffusion models without task-specific retraining. In particular, SyncSDE (Lee et al., 2025) enforces consistency in observed regions while leaving

unobserved regions unconstrained, assuming that realistic completions emerge from contextual alignment. However, its performance remains limited, where it often fails to produce plausible results.

To address this limitation, we propose *Accelerated Likelihood Maximization (ALM)*, a novel training-free sampling strategy for versatile content generation. Unlike prior diffusion synchronization methods that restrict guidance only to observed regions and rely on the implicit assumption that realistic completions will naturally emerge through the diffusion reverse process, ALM explicitly optimizes the unobserved variables during diffusion sampling. In our formulation, these unobserved regions are treated as optimization targets whose likelihood is directly maximized with respect to the observed context. This joint treatment allows the model to enforce local consistency while also aligning the overall sample with the global data distribution, ensuring the results are realistic and semantically coherent.

By explicitly modeling likelihood maximization, ALM generates outputs that are both visually plausible and strongly consistent with the input content across domains. Beyond improving sample quality, ALM is inherently training-free and modality-agnostic, making it broadly applicable without domain-specific retraining. The framework naturally extends to a wide range of tasks—including image inpainting, wide image generation, long video synthesis, and 3D human motion inpainting—while maintaining efficiency through an acceleration mechanism that approximates iterative optimization in a single step. As a result, ALM not only addresses the key shortcomings of prior synchronization-based approaches but also establishes a general paradigm for versatile content generation. Our contributions can be summarized as follows:

- We introduce a novel sampling mechanism, ALM, that explicitly models unobserved regions during diffusion sampling, addressing a central limitation of prior synchronization-based methods.
- ALM is a fully training-free sampling algorithm that requires no task-specific datasets or retraining and can be directly applied to a wide range of pretrained generative models.
- ALM exhibits strong robustness to hyperparameter choices and integrates an acceleration strategy that significantly reduces runtime while maintaining high-fidelity results.
- We demonstrate the versatility and effectiveness of ALM across images, videos, and 3D human motion, achieving state-of-the-art performance even compared to training-based baselines.

## 2 RELATED WORKS

**Training-based approaches.** Several methods require per-task training to address specific subtasks of versatile content generation. For image inpainting, BrushNet (Ju et al., 2024) presents a plug-and-play dual-branch diffusion architecture that separately processes masked image features from diffusion latents. Similarly, PowerPaint (Zhuang et al., 2024) introduces a unified framework with learnable task prompts, allowing a model to handle diverse inpainting challenges within the image domain. Beyond images, CondMDI (Cohan et al., 2024) extends diffusion models to 3D human motion (Tevet et al., 2023) to perform human motion completion from partially observed keyframes, generating coherent and diverse motion sequences. It employs a U-Net-based (Ronneberger et al., 2015) motion diffusion model with randomly sampled keyframes. While these methods achieve strong performance on their specific tasks, their reliance on extensive, task-specific training limits their scalability and generalization across diverse domains.

**Training-free methods.** To overcome the high computation cost required for training-based methods, several task-specific training-free approaches have been proposed. HD-Painter (Manukyan et al., 2025) introduces prompt-aware attention and reweighted attention score guidance to guide the reverse diffusion process, combined with a tailored super-resolution module and Poisson blending (Pérez et al., 2023). Blended latent diffusion (Avrahami et al., 2023) performs cutmix (Yun et al., 2019) operations between foreground and noisy background latents at each denoising step, strictly preserving background through blending operation. Reconstruction guidance (Ho et al., 2022), originally proposed for long video generation, enforces consistency with observed frames during denoising of the unobserved region using L2 loss. This strategy can be extended to other modalities, such as 3D human motion, as discussed in CondMDI (Cohan et al., 2024).

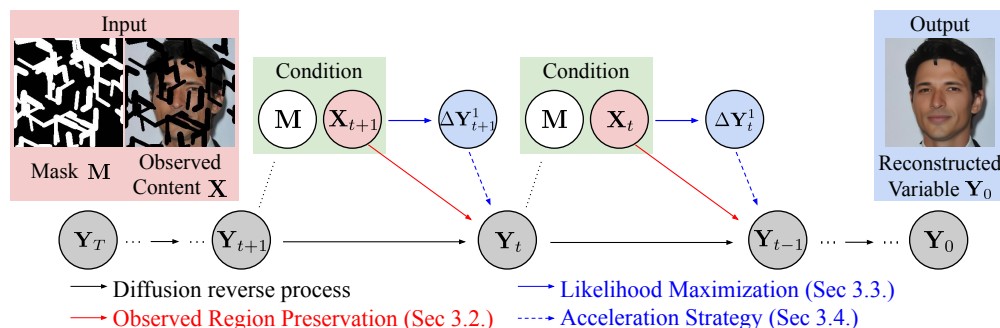

Figure 1: Overview of the proposed method. ALM aims to generate versatile content via reconstructing the unobserved variable.

**Diffusion synchronization.** Diffusion synchronization methods propose customized strategies to model the correlations between different diffusion trajectories for versatile content generation. For instance, SyncTweedies (Kim et al., 2024a) evaluate 60 synchronization strategies and shows that the averaging in the pixel domain using Tweedie's formula (Stein, 1981) yields the better result, though its effectiveness relies largely on heuristics without a clear mathematical explanation. SyncSDE (Lee et al., 2025) addresses this gap by formulating the posterior distribution of the observed content given unobserved region. However, its performance remains limited, as it does not explicitly optimize the unobserved region and instead relies solely on guidance from the observed region during the diffusion sampling. We further analyze its limitations and describe our approach to overcoming them in the following section.

## 3 PROPOSED METHOD

### 3.1 OVERVIEW

We aim to generate versatile content through diffusion-based (Ho et al., 2020; Song et al., 2021a;b) inpainting and outpainting in a training-free manner, where the unobserved variables are sampled while conditioning on the given observed content. We denote the observed content as $\mathbf{X}$ and the binary mask indicating the unobserved regions as $\mathbf{M}$. At diffusion timestep $t$, the noisy observed content is represented as $\mathbf{X}_t$, while the unobserved variable sampled by our method is written as $\mathbf{Y}_t$. We further define the blended content $\mathbf{E}_t$ as $\mathbf{E}_t = \mathbf{X}_t + \mathbf{Y}_t \odot \mathbf{M}$. During diffusion reverse process, we update the unobserved variable $\mathbf{Y}_t$ by modifying the original DDIM (Song et al., 2021a) sampling equation as follows:

$$\mathbf{Y}_{t-1} = \sqrt{\alpha_{t-1}} \left( \frac{\mathbf{Y}_t - \sqrt{1-\alpha_t}\epsilon_\theta(\mathbf{Y}_t, t, \mathbf{c})}{\sqrt{\alpha_t}} \right) + \sqrt{1-\alpha_{t-1}}\epsilon_\theta(\mathbf{Y}_t, t, \mathbf{c})$$
$$+ w_1(\mathbf{1} - \mathbf{M}) \odot (\mathbf{X}_t - \mathbf{Y}_t) + \mathbf{M} \odot (w_2\epsilon_\theta(\mathbf{Y}_t, t, \mathbf{c}) - w_3\epsilon_\theta(\mathbf{E}_t, t, \mathbf{c})), \quad (1)$$

where $\epsilon_\theta$ denotes the pretrained noise prediction network of diffusion model, $\mathbf{c}$ is the conditioning variable (*e.g.* text embedding), and $w_1, w_2, w_3$ are tunable hyperparameters, We briefly introduce our method in this section, and attach the full derivation in the Appendix.

### 3.2 PRELIMINARY: OBSERVED REGION PRESERVATION VIA SYNCSDE

A representative approach tackling training-free versatile content generation is diffusion synchronization (Kim et al., 2024a; Lee et al., 2025). SyncSDE, which provides a probabilistic explanation of why diffusion synchronization works, models the conditional probability of $\mathbf{X}_t$ given $\mathbf{Y}_t, \mathbf{c}$ as:

$$p(\mathbf{X}_t \mid \mathbf{Y}_t, \mathbf{c}) := p(\mathbf{X}_t \mid \mathbf{Y}_t) \sim \mathcal{N}(\mathbf{Y}_t, w_1(1-\alpha_t)(\mathbf{1} - \bar{\mathbf{M}})^{-1}), \quad (2)$$

with a hyperparameter $w_1$ and a diagonal precision matrix $\bar{\mathbf{M}}$, where observed and unobserved entries are set to 0 and 1, respectively. This conditional score is then substituted into the diffusion reverse

process, yielding the update rule:

$$\mathbf{Y}_{t-1} = \sqrt{\alpha_{t-1}} \left( \frac{\mathbf{Y}_t - \sqrt{1-\alpha_t}\epsilon_\theta(\mathbf{Y}_t, t, \mathbf{c})}{\sqrt{\alpha_t}} \right) + \sqrt{1-\alpha_{t-1}}\epsilon_\theta(\mathbf{Y}_t, t, \mathbf{c})$$
$$+ w_1(\mathbf{1} - \mathbf{M}) \odot (\mathbf{X}_t - \mathbf{Y}_t), \tag{3}$$

where the effect of $\gamma_t$ in the last term is absorbed into the value of $w_1$.

### 3.3 Unobserved region optimization via likelihood maximization

Despite the synchronization strategy discussed in Sec. 3.2, it often yields suboptimal results. Our analysis suggests that the guidance mechanism derived in Eq. 3 focuses solely on optimizing the observed region, $(\mathbf{1} - \mathbf{M}) \odot \mathbf{Y}_t$, without explicitly providing any information for the unobserved region, $\mathbf{M} \odot \mathbf{Y}_t$. In other words, SyncSDE (Lee et al., 2025) does not guarantee that the unobserved region will be harmonized with the observed content; instead, it just assumes that synchronization will naturally produce a plausible outcome. As shown in Figure 3 (w/o ALM column), it often fails to generate coherent outputs, where the unobserved regions frequently contain inconsistent or arbitrary content.

Based on the above analysis, we aim to optimize not only the observed region but also the unobserved region of $\mathbf{Y}_t$ by imposing a novel sampling strategy. Our method builds upon the key philosophy of SyncSDE, which guides the reverse diffusion process with a conditional score function to preserve the observed region. At each diffusion timestep $t$, we introduce an additional term $\Delta\mathbf{Y}_t$, which is added into the update rule of Eq. 3. We design $\Delta\mathbf{Y}_t = \sum_{i=1}^{N} \Delta\mathbf{Y}_t^i$, where the sequence of $\{\Delta\mathbf{Y}_t^1, \Delta\mathbf{Y}_t^2, \cdots, \Delta\mathbf{Y}_t^N\}$ is constructed to iteratively minimize the following objective:

$$-\lambda_1 \log p(\mathbf{X}_t, \mathbf{M} \mid \mathbf{Y}_t^i + \mathbf{M} \odot \Delta\mathbf{Y}_t^i, \mathbf{c}) - \lambda_2 \log p(\mathbf{X}_t, \mathbf{M}, \mathbf{Y}_t^i + \mathbf{M} \odot \Delta\mathbf{Y}_t^i \mid \mathbf{c}), \tag{4}$$

with $\lambda_1$ and $\lambda_2$ being scalar hyperparameters ($\lambda_1 > \lambda_2$). Note that $\mathbf{Y}_t^i = \mathbf{Y}_t^{i-1} + \mathbf{M} \odot \Delta\mathbf{Y}_t^{i-1}$, and the initial values are set as $\mathbf{Y}_t^1 = \mathbf{Y}_t$ and $\{\Delta\mathbf{Y}_t^i\}_{i=1}^N = \{\mathbf{0}\}_{i=1}^N$. We distinguish between the roles of conditional and joint likelihoods presented in Eq. 4. The conditional likelihood encourages contextual consistency by aligning the unobserved region with the observed region, whereas the joint likelihood enforces that the blended content lies within the manifold of data distribution, thereby harmonizing both regions. The coefficients $\lambda_1$ and $\lambda_2$ act as weights in a composite energy function (Song et al., 2021b), allowing adaptive balancing between two terms for better performance.

We define $f(\Delta\mathbf{Y}_t^i)$ as the objective defined in Eq. 4. By assuming $|\Delta\mathbf{Y}_t^i| \ll 1$, we apply a Taylor expansion around $\mathbf{0}$. Then by taking a gradient descent step on $\Delta\mathbf{Y}_t$ with step size 1, we obtain:

$$\Delta\mathbf{Y}_t^i = \mathbf{M} \odot (\lambda_1 \nabla_{\mathbf{Y}_t^i} \log p(\mathbf{X}_t, \mathbf{M} \mid \mathbf{Y}_t^i, \mathbf{c}) + \lambda_2 \nabla_{\mathbf{Y}_t^i} \log p(\mathbf{X}_t, \mathbf{M}, \mathbf{Y}_t^i \mid \mathbf{c})). \tag{5}$$

Using the Bayes rule, we factorize the conditional log-likelihood term into $p(\mathbf{X}_t, \mathbf{M}, \mathbf{Y}_t^i \mid \mathbf{c})$ and $p(\mathbf{Y}_t^i \mid \mathbf{c})$. Following Song et al. (2021b), the second term is calculated using the pretrained diffusion model:

$$\nabla_{\mathbf{Y}_t^i} \log p(\mathbf{Y}_t^i \mid \mathbf{c}) \simeq -\frac{1}{\sqrt{1-\alpha_t}}\epsilon_\theta(\mathbf{Y}_t^i, t, \mathbf{c}) \tag{6}$$

For the first term, we assume that $p(\mathbf{X}_t, \mathbf{M}, \mathbf{Y}_t^i \mid \mathbf{c}) \simeq p(\mathbf{E}_t^i \mid \mathbf{c})$, yielding

$$\nabla_{\mathbf{Y}_t^i} \log p(\mathbf{E}_t^i \mid \mathbf{c}) = \nabla_{\mathbf{E}_t^i} \log p(\mathbf{E}_t^i \mid \mathbf{c}) \odot \mathbf{M} \simeq -\frac{1}{\sqrt{1-\alpha_t}}\epsilon_\theta(\mathbf{E}_t^i, t, \mathbf{c}) \odot \mathbf{M}. \tag{7}$$

Putting these together, the closed form formula for $\Delta\mathbf{Y}_t$ becomes

$$\Delta\mathbf{Y}_t^i = \mathbf{M} \odot (\lambda_1(\epsilon_\theta(\mathbf{Y}_t^i, t, \mathbf{c}) - \epsilon_\theta(\mathbf{E}_t^i, t, \mathbf{c})) - \lambda_2\epsilon_\theta(\mathbf{E}_t^i, t, \mathbf{c})). \tag{8}$$

### 3.4 Acceleration strategy

From Eq. 8, we can choose $\lambda_1$ and $\lambda_2$ such that each $\Delta\mathbf{Y}_t^i$ remains sufficiently small, then perform $N$ sequential iterations at every diffusion timestep. However, this iterative process is computationally expensive, since its runtime grows linearly with $N$. To address this, we adopt a *one-step approximation* strategy, which collapses the effect of multiple small updates into a single large step. For the rest of the derivation, we denote $\mathbf{Y}_t^i = \mathbf{Y}_t^{i-1} + \Delta\mathbf{Y}_t^{i-1}$ by definition of $\Delta\mathbf{Y}_t^{i-1}$.

**Claim 1.** $\Delta \mathbf{Y}_t^i$ is small enough for all $1 \le i \le N$. *i.e.* $\lambda_1$ and $\lambda_2$ are chosen such that $|\Delta \mathbf{Y}_t^i| \ll 1$.

**Claim 2.** The noise prediction network $\epsilon_\theta$ of the pretrained diffusion model is $L$-Lipschitz (Karras et al., 2022; Kim et al., 2024b).

Using these claims, we analyze the difference between $\Delta \mathbf{Y}_t^i$ and $\Delta \mathbf{Y}_t^{i+1}$:

$$\|\Delta \mathbf{Y}_t^{i+1} - \Delta \mathbf{Y}_t^i\|$$
$$\le \lambda_1 \|\epsilon_\theta(\mathbf{Y}_t^i + \Delta \mathbf{Y}_t^i, t, \mathbf{c}) - \epsilon_\theta(\mathbf{Y}_t^i, t, \mathbf{c})\| + (\lambda_1 + \lambda_2)\|\epsilon_\theta(\mathbf{E}_t^i + \Delta \mathbf{Y}_t^i, t, \mathbf{c}) - \epsilon_\theta(\mathbf{E}_t^i, t, \mathbf{c})\|$$
$$\le L(2\lambda_1 + \lambda_2)\|\Delta \mathbf{Y}_t^i\| = O(\Delta \mathbf{Y}_t^i)$$

From Claim 1, it follows that $\Delta \mathbf{Y}_t^{i+1} \simeq \Delta \mathbf{Y}_t^i$ for all $i$. Therefore, we approximate the iterative update with a 1-step approximation as follows:

$$\Delta \mathbf{Y}_t \simeq N\Delta \mathbf{Y}_t^1 = \mathbf{M} \odot (w_2 \epsilon_\theta(\mathbf{Y}_t, t, \mathbf{c}) - w_3 \epsilon_\theta(\mathbf{E}_t, t, \mathbf{c})), \tag{9}$$

where we define $w_2 = N\lambda_1$ and $w_3 = N(\lambda_1 - \lambda_2)$. In practice, we empirically apply a decaying schedule to these hyperparameters, defined as:

$$w_i = \sigma_t w_i^{\text{init}}, \quad \sigma_t = \sqrt{\frac{1 - \alpha_{t-1}}{1 - \alpha_t}} \sqrt{1 - \frac{\alpha_t}{\alpha_{t-1}}} \tag{10}$$

where $\sigma_t$ follows the same definition as in DDPM (Ho et al., 2020). We further demonstrate that the proposed acceleration strategy dramatically reduces the required runtime while maintaining the fidelity in Section 4.2.1 and Appendix F.

## 4 EXPERIMENTS

### 4.1 IMPLEMENTATION DETAILS

We implement our method based on PyTorch (Paszke et al., 2019). For the image and video domains, we use a DDIM (Song et al., 2021a) sampler with 50 timesteps, while for the 3D human motion domain we follow the CondMDI setup, which employs a DDPM (Ho et al., 2020) sampler with 1,000 timesteps. To ensure accurate gradient estimation of log-likelihood in Eq. 6 and Eq. 7, we do not employ classifier-free guidance (Ho & Salimans, 2021) during the calculation of Eq. 8. However, for fair comparison with baselines, we still apply classifier-free guidance in the reverse diffusion process of Eq. 3, consistent with all baselines. For SyncSDE (Lee et al., 2025), since the official codebase does not support image inpainting scenarios, we reproduced it.

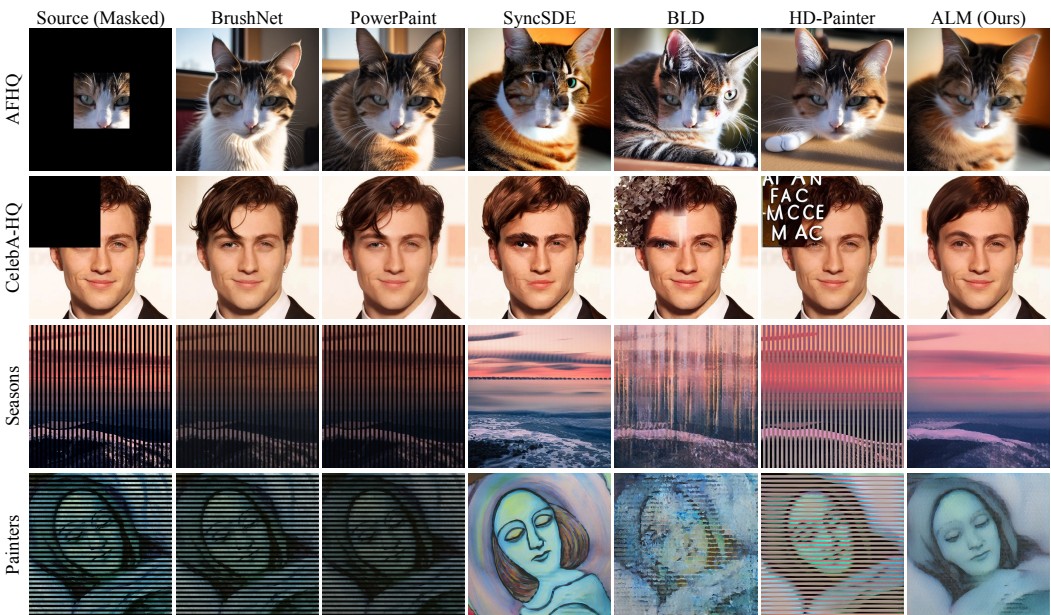

Figure 2: Qualitative comparison of our method against state-of-the-art image inpainting methods (Ju et al., 2024; Zhuang et al., 2024; Lee et al., 2025; Avrahami et al., 2023; Manukyan et al., 2025) using diverse datasets. ALM shows superior performance compared to baselines.

## 4.2 VERSATILE CONTENT GENERATION

We comprehensively evaluate our approach on both inpainting and outpainting across diverse data domains, highlighting its capability for versatile content generation. Specifically, we assess image inpainting in Sec. 4.2.1, wide image generation through outpainting in Sec. 4.2.2. Beyond the image domain, we extend our framework to human motion in Sec. 4.2.3, and further explore its applicability to long video generation in Sec. 4.2.4.

For each table, we **bold** and underline the best and second-best results. The overall ranking ('Rank') is calculated by first ranking each metric individually, averaging the ranks across various metrics, and then ranking these averaged scores. For additional qualitative results, please refer to our Appendix B and Project page submission.

### 4.2.1 IMAGE INPAINTING

Table 1: Quantitative comparison on image inpainting task with state-of-the-art methods (Ju et al., 2024; Zhuang et al., 2024; Lee et al., 2025; Avrahami et al., 2023; Manukyan et al., 2025) using the AFHQ (Choi et al., 2020), CelebA-HQ (Karras et al., 2018), Seasons and Painters (Anoosheh et al., 2018) dataset. Methods with * and † denotes results obtained using pixel-level blending and super-resolution, respectively.

| Method | Training-free | AFHQ | | | | | CelebA-HQ | | | | |
|---|---|---|---|---|---|---|---|---|---|---|---|
| | | LPIPS (↓) | MSE (↓) | SSIM (↑) | CS (↑) | Rank (↓) | LPIPS (↓) | MSE (↓) | SSIM (↑) | CS (↑) | Rank (↓) |
| BrushNet | N | 0.434 | 0.216 | 0.494 | **29.46** | 6 | 0.369 | 0.195 | 0.558 | 27.01 | 4 |
| PowerPaint | N | 0.428 | 0.217 | 0.502 | 29.26 | 5 | 0.365 | 0.203 | 0.568 | 26.84 | 4 |
| SyncSDE | Y | 0.414 | 0.172 | 0.552 | 28.80 | 2 | 0.390 | 0.159 | 0.588 | 28.18 | 3 |
| BLD | Y | 0.421 | 0.149 | 0.520 | 28.71 | 3 | 0.372 | 0.130 | 0.611 | **28.64** | 2 |
| HD-Painter | Y | 0.398 | 0.146 | 0.498 | 28.28 | 3 | 0.374 | 0.146 | 0.550 | 26.80 | 6 |
| ALM (Ours) | Y | **0.391** | **0.142** | **0.591** | 29.00 | 1 | **0.342** | **0.125** | **0.644** | 28.01 | 1 |
| BrushNet* | N | 0.389 | 0.201 | 0.616 | 28.91 | 2 | 0.335 | 0.183 | 0.645 | 26.82 | 2 |
| HD-Painter*† | Y | 0.368 | 0.136 | 0.610 | 28.46 | 2 | 0.355 | 0.140 | 0.619 | 27.48 | 2 |
| ALM* (Ours) | Y | **0.321** | **0.125** | **0.709** | **29.11** | 1 | **0.297** | **0.112** | **0.723** | **27.70** | 1 |

| Method | Training-free | Seasons | | | | | Painters | | | | |
|---|---|---|---|---|---|---|---|---|---|---|---|
| | | LPIPS (↓) | MSE (↓) | SSIM (↑) | CS (↑) | Rank (↓) | LPIPS (↓) | MSE (↓) | SSIM (↑) | CS (↑) | Rank (↓) |
| BrushNet | N | 0.474 | 0.229 | 0.405 | 26.38 | 5 | 0.504 | 0.237 | 0.377 | 26.69 | 5 |
| PowerPaint | N | 0.483 | 0.249 | 0.403 | 26.22 | 6 | 0.523 | 0.247 | 0.379 | 26.95 | 6 |
| SyncSDE | Y | 0.483 | 0.197 | 0.435 | **27.75** | 2 | 0.546 | 0.194 | 0.408 | **28.18** | 3 |
| BLD | Y | 0.458 | 0.170 | 0.422 | 26.54 | 4 | 0.497 | 0.163 | 0.392 | 27.37 | 2 |
| HD-Painter | Y | **0.435** | 0.163 | 0.423 | 25.72 | 2 | **0.485** | 0.200 | 0.387 | 26.31 | 4 |
| ALM (Ours) | Y | 0.456 | **0.162** | **0.471** | 26.60 | 1 | 0.504 | **0.153** | **0.442** | 27.26 | 1 |
| BrushNet* | N | 0.396 | 0.202 | 0.599 | 26.04 | 2 | 0.423 | 0.216 | 0.581 | 26.08 | 2 |
| HD-Painter*† | Y | 0.382 | 0.142 | 0.597 | 25.87 | 2 | 0.434 | 0.184 | 0.573 | 26.37 | 2 |
| ALM* (Ours) | Y | **0.350** | **0.132** | **0.675** | **26.56** | 1 | **0.361** | **0.127** | **0.680** | **26.58** | 1 |

**Comparison with State-of-the-Art methods.** We conducted a comprehensive comparison of our method against a wide range of state-of-the-art inpainting techniques. The baselines include training-based methods like BrushNet (Ju et al., 2024), PowerPaint (Zhuang et al., 2024) and training-free method such as SyncSDE (Lee et al., 2025), Blended Latent Diffusion (Avrahami et al., 2023), and HD-Painter (Manukyan et al., 2025) using the pretrained Stable Diffusion. By comparing our method against SyncSDE (Lee et al., 2025), we provide strong supporting evidences for the analysis represented in Sec 3.3. For our evaluation, we used four distinct datasets: AFHQ (Choi et al., 2020), CelebA-HQ (Karras et al., 2018), Seasons, and Painters (Anoosheh et al., 2018). From each dataset, we randomly sample 1,000 image-mask pairs to construct the test set. We measure the performance using four commonly adopted metrics; LPIPS (Zhang et al., 2018), MSE, SSIM, and CLIP Score (CS) (Radford et al., 2021). Since both ALM and SyncSDE require the sequence $\{\mathbf{X}_t\}_{t=0}^{T}$, we apply DDIM (Song et al., 2021a) inversion as a preprocessing.

As summarized in Table 1, our method, ALM, demonstrates outstanding performance across all baselines. Notably, it consistently outperforms both training-free and even training-based methods, regardless of the dataset. Figure 2 visualizes the qualitative comparisons, where our method consistently delivers superior visual quality. Our method also shows robust performance with diverse and complex mask shapes, demonstrating its generalizability. In addition, we emphasize that ALM is robust to variations in hyperparameters choices, consistently achieving strong performance across configurations, as detailed in Appendix D. We further compare ALM with Stable Diffusion Inpainting in Appendix B.1 and show that our method shows superior results. Finally, Appendix E demonstrates that ALM effectively handles images conditioned on long and complex text prompts, highlighting its performance across challenging scenarios.

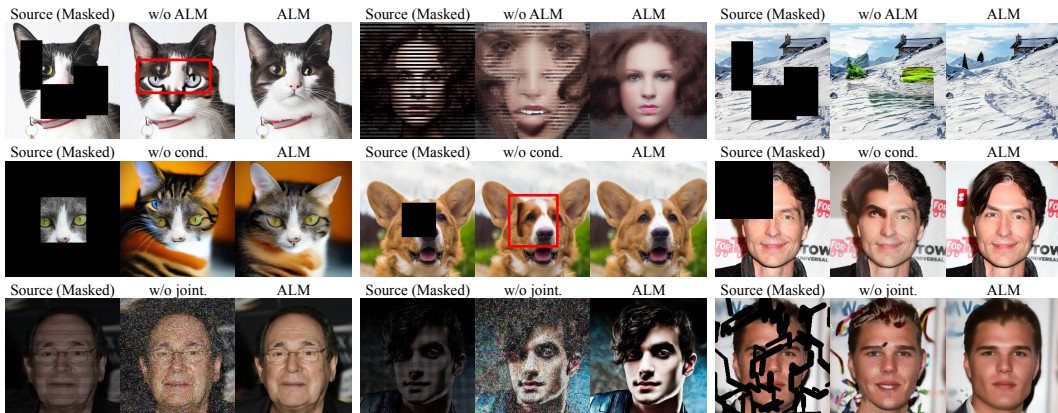

Figure 3: Ablation study results. We show the effectiveness of ALM (Eq. 8), conditional likelihood term (Eq. 4), and joint likelihood term (Eq. 4) in each row, respectively.

**Analyzing the effect of the ALM.** We analyze the effect of ALM by comparing with a baseline that applies only the update rule in Eq. 3. As shown in Table 2, the proposed method shows consistent improvements across diverse datasets, with clear quantitative gains. Moreover, as illustrated in the first row of Figure 3, it effectively mitigates a key limitation of the existing diffusion synchronization framework (Lee et al., 2025), which tends to overemphasize observed regions. In contrast, by explicitly optimizing conditional and joint likelihood terms for unobserved regions, our approach produces globally coherent samples that align well with the given context.

We further assess the effectiveness of the conditional and joint likelihood terms in our ALM's optimization objective (Eq. 4). As illustrated in the second and third rows of Figure 3, both terms are crucial for generating high-quality outputs. Specifically, the conditional likelihood term is more effective with a pretrained conditional model (e.g., Stable Diffusion), while the joint term has a greater impact on unconditional diffusion models. This demonstrates that incorporating both terms enables ALM to generalize effectively across diverse diffusion models.

Table 2: Quantitative ablation results evaluating the effect of ALM in image inpainting.

| Method | AFHQ | | | | CelebA-HQ | | | | Seasons | | | |
|---|---|---|---|---|---|---|---|---|---|---|---|---|
| | LPIPS (↓) | MSE (↓) | SSIM (↑) | CS (↑) | LPIPS (↓) | MSE (↓) | SSIM (↑) | CS (↑) | LPIPS (↓) | MSE (↓) | SSIM (↑) | CS (↑) |
| w/o ALM (Eq. 3) | 0.401 | 0.169 | 0.559 | 28.88 | 0.384 | 0.161 | 0.594 | **28.29** | 0.470 | 0.189 | 0.441 | **27.47** |
| ALM (Ours, Eq. 1) | **0.391** | **0.142** | **0.591** | **29.00** | **0.342** | **0.125** | **0.644** | 28.01 | **0.456** | **0.162** | **0.471** | 26.60 |

**Ablation on acceleration strategy.** We provide qualitative comparisons with and without the acceleration strategy in Figure 4 The visual quality remains consistent in both settings, showing that our acceleration strategy reduces computational cost without sacrificing performance. Further runtime analysis is provided in Appendix F.

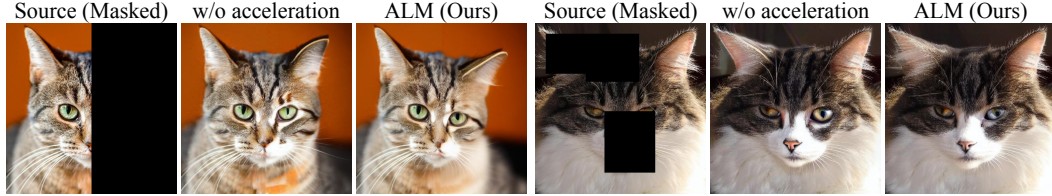

Figure 4: Qualitative comparisons with and without the acceleration strategy. We emphasize that visual quality remains consistent across both setups.

**Experiments across different backbones.** To validate the robustness of ALM with respect to the underlying diffusion backbone, we conduct experiments with multiple diffusion models. We first adopt an unconditional diffusion model trained on CelebA-HQ (Karras et al., 2018) using the official checkpoint from RePaint (Lugmayr et al., 2022). As shown in Figure 5 (top), our method produces plausible inpainting results across diverse scenarios, including challenging cases such as aliasing-pattern and super-resolution masks.

We further evaluate using Stable Diffusion XL (Podell et al., 2024) architecture. As illustrated in Figure 5 (bottom), our method again delivers high-quality inpainting results. These demonstrate that

ALM is general and architecture-robust: across three different diffusion backbones–an unconditional model, Stable Diffusion, and Stable Diffusion XL–our method consistently shows high-quality performance, effectively broadening the applicability of modern foundation models beyond standard generation tasks. We also compare ALM with SDXL-Inpainting in Appendix B.1 and show that our method achieves comparable performance.

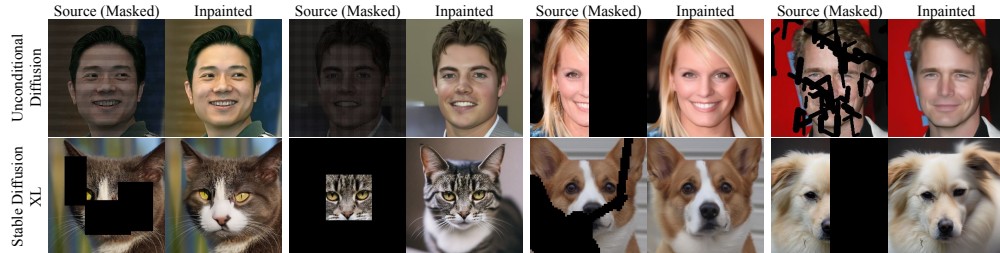

Figure 5: (Top) Qualitative results of image inpainting using the unconditional diffusion model (Lugmayr et al., 2022) trained on CelebA-HQ (Karras et al., 2018) dataset. (Bottom) Inpainted images sampled from AFHQ (Choi et al., 2020) dataset using the pretrained SDXL (Podell et al., 2024).

**Analysis on computational cost.** We quantitatively evaluate the computational cost of ALM in comparison to baseline methods. Table 3 reports the required GPU memory (GB) and runtime (s) for generating a single image. Overall, ALM is fully training-free, and its computational cost remains on par with existing training-free baselines. With the results in Table 1, these findings highlight that ALM achieves the best trade-off between performance and computational efficiency among the various methods.

Table 3: Quantitative computational cost analysis across diverse baselines.

| Method | Additional training cost | GPU Memory (GB) ($\downarrow$) | Runtime (s) ($\downarrow$) | LPIPS ($\downarrow$) |
|---|---|---|---|---|
| BrushNet | 3 days with 8 V100 GPUs | 4.73 | 3.189 | 0.434 |
| PowerPaint | requires 8 A100 GPUs | 5.54 | 4.089 | 0.428 |
| SyncSDE | - | 4.98 | 6.734 | 0.414 |
| BLD | - | **3.39** | **2.615** | 0.421 |
| HD-Painter | - | 29.07 | 38.840 | 0.398 |
| ALM (Ours) | - | 4.98 | 8.584 | **0.391** |
| ALM w/o inversion | - | 4.98 | 3.850 | 0.606 |

### 4.2.2 WIDE IMAGE GENERATION VIA OUTPAINTING

Beyond image inpainting, our approach also naturally extends to the outpainting task, enabling the synthesis of wide, high-resolution images. We employ an autoregressive image outpainting strategy to generate wide images. Starting from an $512 \times 512$ patch generated with the pretrained Stable Diffusion (Rombach et al., 2022), subsequent overlapping patches are iteratively synthesized via outpainting. The patches are overlapped such that the $i$-th patch is placed on top of the $(i-1)$-th one and decoded using the pretrained VAE (Kingma & Welling, 2013) decoder of Stable Diffusion. With a stride of 384 pixels, we generate seven patches in total, resulting in a $2048 \times 512$ resolution image. We compare our method against state-of-the-art diffusion synchronization approaches, including SyncTweedies (Kim et al., 2024a) and SyncSDE (Lee et al., 2025). We report FID (Heusel et al., 2017), KID (Bińkowski et al., 2018), Aesthetic Score (Schuhmann et al., 2022) to evaluate the fidelity of the generated image, and CLIP Score (Radford et al., 2021) to quantify text-image alignment. As shown in Table 4 and Figure 6, our methods achieves outstanding performance compared to the baselines. Since our approach is training-free and does not rely on specific mask priors, our approach exhibits high versatility, seamlessly generalizing to outpainting tasks.

Table 4: Quantitative comparison on wide image generation task with state-of-the-art methods (Kim et al., 2024a; Lee et al., 2025). KID (Bińkowski et al., 2018) metric is scaled by $10^3$.

| Method | FID ($\downarrow$) | KID ($\downarrow$) | Aesthetic Score ($\uparrow$) | CLIP Score ($\uparrow$) | Rank ($\downarrow$) |
|---|---|---|---|---|---|
| SyncTweedies | 108.73 | 61.08 | 6.065 | 33.16 | 3 |
| SyncSDE | 106.81 | 58.79 | 6.049 | **33.47** | 2 |
| ALM (Ours) | **102.30** | **46.01** | **6.086** | 33.27 | **1** |

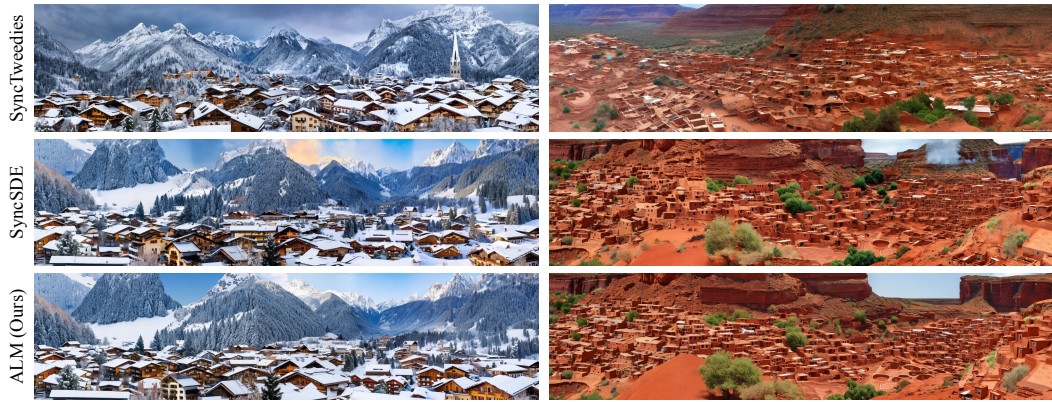

"Alpine village, snow-covered rooftops, nestled between majestic peaks—a picture-perfect scene of winter tranquility"

"Nestled in a canyon, a pueblo village stands against the red earth"

Figure 6: Qualitative comparison of our method against state-of-the-art wide image generation methods (Kim et al., 2024a; Lee et al., 2025). While baseline-generated images often exhibit artifacts and blur, ALM produces wide images with higher fidelity and improved global coherence. For example, (Left) SyncSDE exhibits noticeable color inconsistencies and edge artifacts across patch boundaries, whereas ALM clearly alleviates these issues. (Right) SyncTweedies produces blurry regions on the left side of the image, and SyncSDE shows structural inconsistency in the upper-right area. In contrast, ALM generates a wide image with neither blurred nor inconsistent regions.

### 4.2.3 HUMAN MOTION INPAINTING

We demonstrate the versatility of our method by extending its application beyond images to 3D human motion data. Specifically, we tackle the task of human motion inpainting, where the goal is to reconstruct missing parts of a motion sequence. We evaluated the performance across two distinct scenarios: the "first-half prediction," where the task is to predict the initial part of a sequence given only the latter half, and the "middle-half prediction," where the model must fill in the central portion given the first and last quarters. We utilized a U-Net-based pre-trained diffusion model (Karunratanakul et al., 2023) for text-to-motion synthesis.

We compare our method against training-based method CondMDI (Cohan et al., 2024), and training-free methods like Reconstruction Guidance (Ho et al., 2022) and its Imputation-based variant (Tevet et al., 2023). For each inpainting scenario, we sample 1,000 motion sequences from the HumanML3D (Guo et al., 2022) dataset and report the average performance over 10 replications. We measure FID (Heusel et al., 2017), Matching Score, R-precision, and Diversity metrics. Table 5 illustrate that the proposed method achieves superior performance with high versatility across various human motion inpainting scenarios. In Figure 7, the given frames are highlighted in orange, while the inpainted frames generated by the model are shown in blue. We further conduct an ablation study, as presented in Table 6, demonstrating that ALM substantially contributes to these performance gains.

First Half          Middle Half

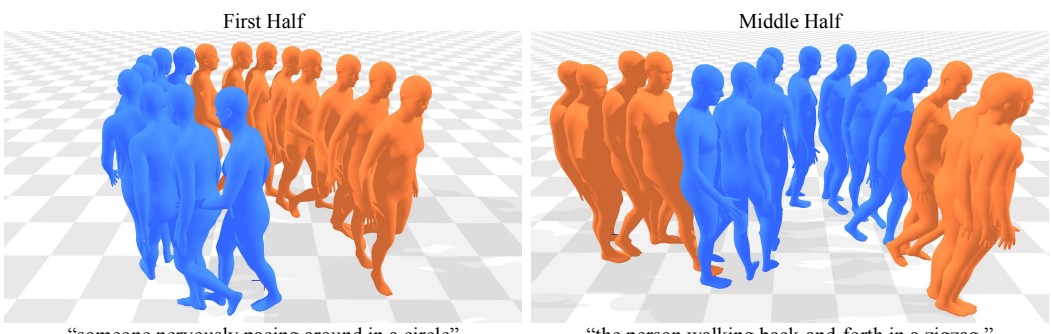

"someone nervously pacing around in a circle"     "the person walking back-and-forth in a zigzag."

Figure 7: Qualitative result on human motion inpainting task.

Table 5: Quantitative comparison on human motion inpainting task with state-of-the-art baselines Cohan et al. (2024); Ho et al. (2022) using the motion sequences sampled from HumanML3D (Guo et al., 2022) dataset. Methods with * indicate results obtained with imputation (Tevet et al., 2023).

| Method | Training-free | First half | | | | | Middle half | | | | |
|---|---|---|---|---|---|---|---|---|---|---|---|
| | | FID (↓) | Match. (↓) | R-Prec. (↑) | Diversity (↑) | Rank (↓) | FID (↓) | Match. (↓) | R-Prec. (↑) | Diversity (↑) | Rank (↓) |
| CondMDI | N | 0.626 | 4.510 | 0.356 | 8.724 | 3 | 0.599 | 4.429 | 0.360 | 8.574 | 3 |
| Recon. Gui. | Y | 11.742 | 5.199 | 0.279 | 6.076 | 5 | 13.113 | 5.124 | 0.295 | 6.018 | 5 |
| Recon. Gui.* | Y | 3.738 | 4.359 | 0.360 | 7.745 | 3 | 4.390 | 4.383 | 0.362 | 7.589 | 4 |
| Ours | Y | **0.346** | 4.112 | **0.400** | **8.926** | **1** | 0.494 | 4.136 | **0.395** | **8.728** | **1** |
| Ours* | Y | 0.503 | **4.098** | 0.398 | 8.804 | 2 | 0.692 | **4.126** | 0.394 | 8.595 | 2 |

Table 6: Quantitative evaluation of ALM's effect on human motion inpainting.

| Method | FID (↓) | Skating Ratio (↓) | Trajectory Error (↓) | Keyframe Error (↓) |
|---|---|---|---|---|
| w/o ALM (Eq. 3) | 0.412 | 0.121 | 0.917 | 0.518 |
| ALM (Ours, Eq. 1) | **0.346** | **0.118** | **0.907** | **0.386** |

#### 4.2.4 LONG VIDEO GENERATION

We extend our versatility into the video domain by generating temporally long sequences. Analogous to the wide image generation task, we produce videos by autoregressively sampling multiple overlapping video patches along the temporal axis. For implementation, we employ the pretrained LaVie (Wang et al., 2025), a diffusion-based text-to-video model that generates $512 \times 320$ resolution videos in the first stage, producing 16 frames from a single text prompt. By setting a temporal stride of 8 and synthesizing a total of 12 patches, we generate a 104-frame video. Consecutive patches are overlaid along the temporal axis and decoded using LaVie's pretrained VAE (Kingma & Welling, 2013) decoder. We visualize the generated long video sequences in Figure 8. As shown, ALM produces visually coherent and semantically consistent sequences, maintaining spatio-temporal continuity. Furthermore, Appendix E demonstrates that ALM remains effective even for videos with dynamic motions, highlighting its robustness in complex scenarios. Lastly, we compare our method against three baselines; FreeNoise (Qiu et al., 2024), SEINE (Chen et al., 2023), and SyncSDE (Lee et al., 2025) and present the results in Appendix G. We use a total of 250 video sequences for evaluation, each containing 104 frames. As shown, our method outperforms the baselines, emphasizing the effectiveness of our method.

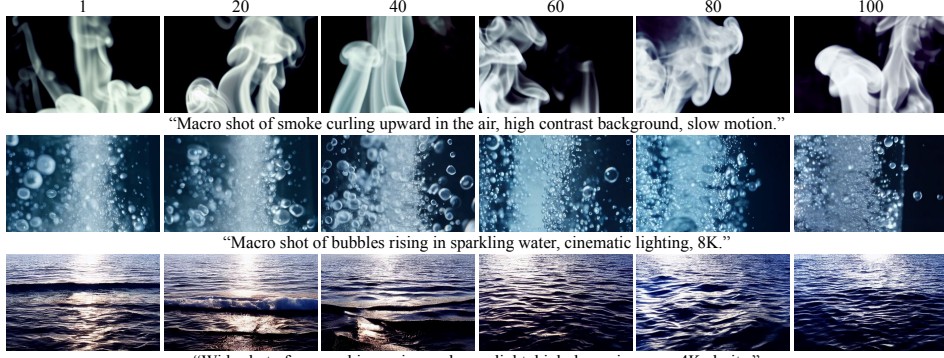

"Macro shot of smoke curling upward in the air, high contrast background, slow motion."

"Macro shot of bubbles rising in sparkling water, cinematic lighting, 8K."

"Wide shot of waves shimmering under sunlight, high dynamic range, 4K clarity."

Figure 8: Qualitative results of long video generation. We use the pretrained LaVie (Wang et al., 2025), which generates 16 frame videos by default, and extend the synthesized videos to 104 frames using ALM.

## 5 CONCLUSION

In this work, we introduce a novel, training-free sampling strategy for diffusion-based versatile content generation. Versatile content generation relies heavily on inpainting and outpainting, where diffusion models often struggle due to their limited generalization capacity, despite the fact that solving these problems gradually expands their applicability. Unlike prior approaches that require per-scenario tuning, our method is entirely training-free and can be generalized across diverse domains, thereby effectively mitigating the inherent limitations of diffusion models. Building on recent advances in diffusion synchronization, we synchronize observed content with unobserved variables by maximizing both joint and conditional likelihoods. Furthermore, we propose a computationally efficient acceleration strategy for likelihood maximization. Experimental results across diverse tasks and modalities demonstrate that our approach achieves state-of-the-art performance in versatile content generation.

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

# A DETAILED DERIVATION OF ALM

## A.1 PRELIMINARY: OBSERVED REGION PRESERVATION VIA SYNCSDE

A representative approach for versatile content generation is diffusion synchronization (Kim et al., 2024a; Lee et al., 2025). SyncSDE, which provides a probabilistic explanation of why diffusion synchronization works, generates content by introducing a conditional probability term that couples different diffusion trajectories. Specifically, it factorizes the conditional score function of the diffusion model used during the sampling of $\mathbf{Y}_t$ as:

$$\nabla_{\mathbf{Y}_t} \log p(\mathbf{Y}_t \mid \mathbf{X}_t, \mathbf{c}) = \nabla_{\mathbf{Y}_t} \log p(\mathbf{Y}_t \mid \mathbf{c}) + \nabla_{\mathbf{Y}_t} \log p(\mathbf{X}_t \mid \mathbf{Y}_t, \mathbf{c}), \tag{11}$$

where the conditional probability of $\mathbf{X}_t$ given $\mathbf{Y}_t, \mathbf{c}$ is modeled as:

$$p(\mathbf{X}_t \mid \mathbf{Y}_t, \mathbf{c}) := p(\mathbf{X}_t \mid \mathbf{Y}_t) \sim \mathcal{N}(\mathbf{Y}_t, w_1(1-\alpha_t)(\mathbf{1}-\bar{\mathbf{M}})^{-1}), \tag{12}$$

with a hyperparameter $w_1$, and a diagonal precision matrix $\bar{\mathbf{M}}$, where the observed and unobserved entries are set to 0 and 1, respectively. This conditional score is then substituted into the diffusion reverse process, yielding the update rule:

$$\mathbf{Y}_{t-1} = \sqrt{\frac{\alpha_{t-1}}{\alpha_t}} \mathbf{Y}_t + (1-\alpha_t)\gamma_t \nabla_{\mathbf{Y}_t} \log p(\mathbf{Y}_t \mid \mathbf{X}_t, \mathbf{c}), \tag{13}$$

where $\gamma_t = \sqrt{\alpha_{t-1}/\alpha_t} - \sqrt{(1-\alpha_{t-1})/(1-\alpha_t)}$. Finally, this leads to the following modified update equation as follows:

$$\mathbf{Y}_{t-1} = \sqrt{\alpha_{t-1}} \left( \frac{\mathbf{Y}_t - \sqrt{1-\alpha_t}\epsilon_\theta(\mathbf{Y}_t, t, \mathbf{c})}{\sqrt{\alpha_t}} \right) + \sqrt{1-\alpha_{t-1}}\epsilon_\theta(\mathbf{Y}_t, t, \mathbf{c})$$
$$+ w_1(\mathbf{1}-\mathbf{M}) \odot (\mathbf{X}_t - \mathbf{Y}_t), \tag{14}$$

where the effect of $\gamma_t$ in the last term is absorbed into the value of $w_1$.

## A.2 UNOBSERVED REGION OPTIMIZATION VIA LIKELIHOOD MAXIMIZATION

Despite the synchronization strategy discussed in Sec. A.1, it often yields suboptimal results. Our analysis suggests that the guidance mechanism derived in Eq. 14 focuses solely on optimizing the observed region, $(\mathbf{1}-\mathbf{M}) \odot \mathbf{Y}_t$, without explicitly providing any information for the unobserved region, $\mathbf{M} \odot \mathbf{Y}_t$. In other words, SyncSDE (Lee et al., 2025) does not guarantee that the unobserved region will be harmonized with the observed content; instead, it just assumes that synchronization will naturally produce a plausible outcome. To validate this analysis, we conduct an experiment on image inpainting using the pretrained Stable Diffusion (Rombach et al., 2022). As shown in Figure 3 (1st row, "w/o ALM" columns), it often fails to generate coherent outputs, where the unobserved regions frequently contain inconsistent or arbitrarily generated content that does not harmonize with the observed region. In contrast, our method successfully synthesizes the unobserved region, with the tailored sampling strategy which we detail in this section.

Based on the above analysis, we aim to optimize not only the observed region but also the unobserved region of $\mathbf{Y}_t$ by imposing a novel sampling strategy. Our method builds upon the key philosophy of SyncSDE, which guides the reverse diffusion process with a conditional score function to preserve the observed region. At each diffusion timestep $t$, we introduce an additional term $\Delta \mathbf{Y}_t$, which is incorporated into the update rule of Eq. 14, to revise the reverse diffusion process as:

$$\mathbf{Y}_{t-1} = \sqrt{\alpha_{t-1}} \left( \frac{\mathbf{Y}_t - \sqrt{1-\alpha_t}\epsilon_\theta(\mathbf{Y}_t, t, \mathbf{c})}{\sqrt{\alpha_t}} \right) + \sqrt{1-\alpha_{t-1}}\epsilon_\theta(\mathbf{Y}_t, t, \mathbf{c})$$
$$+ w_1(\mathbf{1}-\mathbf{M}) \odot (\mathbf{X}_t - \mathbf{Y}_t) + \Delta \mathbf{Y}_t. \tag{15}$$

We design $\Delta \mathbf{Y}_t = \sum_{i=1}^{N} \Delta \mathbf{Y}_t^i$, where the sequence of $\{\Delta \mathbf{Y}_t^1, \Delta \mathbf{Y}_t^2, \cdots, \Delta \mathbf{Y}_t^N\}$ is constructed to iteratively minimize the following objective:

$$-\lambda_1 \log p(\mathbf{X}_t, \mathbf{M} \mid \mathbf{Y}_t^i + \mathbf{M} \odot \Delta \mathbf{Y}_t^i, \mathbf{c}) - \lambda_2 \log p(\mathbf{X}_t, \mathbf{M}, \mathbf{Y}_t^i + \mathbf{M} \odot \Delta \mathbf{Y}_t^i \mid \mathbf{c}), \tag{16}$$

with $\lambda_1$ and $\lambda_2$ being scalar hyperparameters ($\lambda_1 > \lambda_2$). Note that $\mathbf{Y}_t^i = \mathbf{Y}_t^{i-1} + \mathbf{M} \odot \Delta \mathbf{Y}_t^{i-1}$, and the initial values are set as $\mathbf{Y}_t^1 = \mathbf{Y}_t$ and $\{\Delta \mathbf{Y}_t^i\}_{i=1}^{N} = \{\mathbf{0}\}_{i=1}^{N}$. We distinguish between the

roles of conditional and joint likelihoods presented in Eq. 16. The conditional likelihood encourages contextual consistency by aligning the unobserved region with the observed region, whereas the joint likelihood enforces that the blended content lies within the support of full data distribution, thereby harmonizing both regions into a globally realistic sample. This separation enables our method to simultaneously preserve local consistency and guarantee global harmonization. The coefficients $\lambda_1$ and $\lambda_2$ act as weights in a composite energy function (Song et al., 2021b), allowing adaptive balancing between two terms for better performance.

We define $f(\Delta \mathbf{Y}_t^i)$ as the optimization objective defined in Eq. 4. By assuming $|\Delta \mathbf{Y}_t^i| \ll 1$ and applying a Taylor expansion around $\mathbf{0}$, we derive:

$$
\begin{aligned}
f(\Delta \mathbf{Y}_t^i) \simeq & -\lambda_1 (\log p(\mathbf{X}_t, \mathbf{M} \mid \mathbf{Y}_t^i, \mathbf{c}) + (\mathbf{M} \odot \nabla_{\mathbf{Y}_t^i} \log p(\mathbf{X}_t, \mathbf{M} \mid \mathbf{Y}_t^i, \mathbf{c})))^\top \Delta \mathbf{Y}_t^i \\
& -\lambda_2 (\log p(\mathbf{X}_t, \mathbf{M}, \mathbf{Y}_t^i \mid \mathbf{c}) + (\mathbf{M} \odot \nabla_{\mathbf{Y}_t^i} \log p(\mathbf{X}_t, \mathbf{M}, \mathbf{Y}_t^i \mid \mathbf{c})))^\top \Delta \mathbf{Y}_t^i.
\end{aligned} \tag{17}
$$

Taking a gradient descent step with respect to $\Delta \mathbf{Y}_t$ with step size set to 1, we obtain:

$$
\Delta \mathbf{Y}_t^i = \mathbf{M} \odot (\lambda_1 \nabla_{\mathbf{Y}_t^i} \log p(\mathbf{X}_t, \mathbf{M} \mid \mathbf{Y}_t^i, \mathbf{c}) + \lambda_2 \nabla_{\mathbf{Y}_t^i} \log p(\mathbf{X}_t, \mathbf{M}, \mathbf{Y}_t^i \mid \mathbf{c})). \tag{18}
$$

We further factorize the conditional log-likelihood as follows:

$$
\nabla_{\mathbf{Y}_t^i} \log p(\mathbf{X}_t, \mathbf{M} \mid \mathbf{Y}_t^i, \mathbf{c}) = \nabla_{\mathbf{Y}_t^i} \log p(\mathbf{X}_t, \mathbf{M}, \mathbf{Y}_t^i \mid \mathbf{c}) - \nabla_{\mathbf{Y}_t^i} \log p(\mathbf{Y}_t^i \mid \mathbf{c}). \tag{19}
$$

Following Song et al. (2021b), the second term is calculated using the pretrained diffusion model:

$$
\nabla_{\mathbf{Y}_t^i} \log p(\mathbf{Y}_t^i \mid \mathbf{c}) \simeq -\frac{1}{\sqrt{1-\alpha_t}} \epsilon_\theta(\mathbf{Y}_t^i, t, \mathbf{c}) \tag{20}
$$

For the first term, we assume that $p(\mathbf{X}_t, \mathbf{M}, \mathbf{Y}_t^i \mid \mathbf{c}) \simeq p(\mathbf{E}_t^i \mid \mathbf{c})$, yielding

$$
\nabla_{\mathbf{Y}_t^i} \log p(\mathbf{E}_t^i \mid \mathbf{c}) = \nabla_{\mathbf{E}_t^i} \log p(\mathbf{E}_t^i \mid \mathbf{c}) \odot \mathbf{M} \simeq -\frac{1}{\sqrt{1-\alpha_t}} \epsilon_\theta(\mathbf{E}_t^i, t, \mathbf{c}) \odot \mathbf{M}. \tag{21}
$$

Putting these together, the closed form formula for $\Delta \mathbf{Y}_t$ becomes

$$
\Delta \mathbf{Y}_t^i = \mathbf{M} \odot (\lambda_1 (\epsilon_\theta(\mathbf{Y}_t^i, t, \mathbf{c}) - \epsilon_\theta(\mathbf{E}_t^i, t, \mathbf{c})) - \lambda_2 \epsilon_\theta(\mathbf{E}_t^i, t, \mathbf{c})). \tag{22}
$$

### A.3 ACCELERATION STRATEGY

From the relation

$$
|\Delta \mathbf{Y}_t^i| \leq |\lambda_1| \|\epsilon_\theta(\mathbf{Y}_t^i, t, \mathbf{c}) - \epsilon_\theta(\mathbf{E}_t^i, t, \mathbf{c})\| + |\lambda_2| \|\epsilon_\theta(\mathbf{E}_t^i, t, \mathbf{c})\|, \tag{23}
$$

we can choose $\lambda_1$ and $\lambda_2$ such that each $\Delta \mathbf{Y}_t^i$ remains sufficiently small, then perform $N$ sequential iterations at every diffusion timestep. However, this iterative process is computationally expensive, since its runtime grows linearly with $N$. To address this, we adopt a *one-step approximation* strategy, which collapses the effect of multiple small updates into a single large step. Intuitively, instead of gradually refining the unobserved region through lots of iterations, we directly approximate the outcome of the full optimization in a single update, significantly reducing computation time while preserving the intended correction. For the rest of the derivation, we denote $\mathbf{Y}_t^i = \mathbf{Y}_t^{i-1} + \Delta \mathbf{Y}_t^{i-1}$ by definition of $\Delta \mathbf{Y}_t^{i-1}$.

**Claim 1.** $\Delta \mathbf{Y}_t^i$ is small enough for all $1 \leq i \leq N$. *i.e.* $\lambda_1$ and $\lambda_2$ are chosen such that $|\Delta \mathbf{Y}_t^i| \ll 1$.
**Claim 2.** The noise prediction network $\epsilon_\theta$ of the pretrained diffusion model is $L$-Lipschitz (Karras et al., 2022; Kim et al., 2024b).

Using these claims, we analyze the difference between $\Delta \mathbf{Y}_t^i$ and $\Delta \mathbf{Y}_t^{i+1}$:

$$
\begin{aligned}
& \|\Delta \mathbf{Y}_t^{i+1} - \Delta \mathbf{Y}_t^i\| \\
& \quad \leq \|(\lambda_1 (\epsilon_\theta(\mathbf{Y}_t^i + \Delta \mathbf{Y}_t^i, t, \mathbf{c}) - \epsilon_\theta(\mathbf{E}_t^i + \Delta \mathbf{Y}_t^i, t, \mathbf{c})) - \lambda_2 \epsilon_\theta(\mathbf{E}_t^i + \Delta \mathbf{Y}_t^i, t, \mathbf{c})) \\
& \qquad - (\lambda_1 (\epsilon_\theta(\mathbf{Y}_t^i, t, \mathbf{c}) - \epsilon_\theta(\mathbf{E}_t^i, t, \mathbf{c})) - \lambda_2 \epsilon_\theta(\mathbf{E}_t^i, t, \mathbf{c}))\| \\
& \quad = \|\lambda_1 (\epsilon_\theta(\mathbf{Y}_t^i + \Delta \mathbf{Y}_t^i, t, \mathbf{c}) - \epsilon_\theta(\mathbf{Y}_t^i, t, \mathbf{c})) - (\lambda_1 + \lambda_2)(\epsilon_\theta(\mathbf{E}_t^i + \Delta \mathbf{Y}_t^i, t, \mathbf{c}) - \epsilon_\theta(\mathbf{E}_t^i, t, \mathbf{c}))\| \\
& \quad \leq \lambda_1 \|\epsilon_\theta(\mathbf{Y}_t^i + \Delta \mathbf{Y}_t^i, t, \mathbf{c}) - \epsilon_\theta(\mathbf{Y}_t^i, t, \mathbf{c})\| + (\lambda_1 + \lambda_2) \|\epsilon_\theta(\mathbf{E}_t^i + \Delta \mathbf{Y}_t^i, t, \mathbf{c}) - \epsilon_\theta(\mathbf{E}_t^i, t, \mathbf{c})\| \\
& \quad \leq L(2\lambda_1 + \lambda_2) \|\Delta \mathbf{Y}_t^i\| \\
& \quad = O(\Delta \mathbf{Y}_t^i)
\end{aligned} \tag{24}
$$

From Claim 1, it follows that $\Delta \mathbf{Y}_t^{i+1} \simeq \Delta \mathbf{Y}_t^i$ for all $i$. Therefore, we approximate the iterative update with a 1-step approximation as follows:

$$\Delta \mathbf{Y}_t \simeq N \Delta \mathbf{Y}_t^1 = \mathbf{M} \odot \left( N\lambda_1(\epsilon_\theta(\mathbf{Y}_t, t, \mathbf{c}) - \epsilon_\theta(\mathbf{E}_t, t, \mathbf{c})) - N\lambda_2 \epsilon_\theta(\mathbf{E}_t, t, \mathbf{c}) \right)$$
$$= \mathbf{M} \odot \left( w_2 \epsilon_\theta(\mathbf{Y}_t, t, \mathbf{c}) - w_3 \epsilon_\theta(\mathbf{E}_t, t, \mathbf{c}) \right), \qquad (25)$$

where we define $w_2 = N\lambda_1$ and $w_3 = N(\lambda_1 - \lambda_2)$. In practice, we empirically apply a decaying schedule to these hyperparameters, defined as:

$$w_i = \sigma_t w_i^{\text{init}}, \quad \sigma_t = \sqrt{\frac{1 - \alpha_{t-1}}{1 - \alpha_t}} \sqrt{1 - \frac{\alpha_t}{\alpha_{t-1}}} \qquad (26)$$

where $\sigma_t$ follows the same definition as in DDPM (Ho et al., 2020). For all experiments, we set the hyperparameter value to satisfy $w_1^{\text{init}} = w_3^{\text{init}}$.

# B  ADDITIONAL EXPERIMENTAL RESULTS

In this section, we present an additional versatile content generation result with experimental details.

## B.1  IMAGE INPAINTING

We visualize additional image inpainting results using images brought from the AFHQ (Choi et al., 2020), CelebA-HQ (Karras et al., 2018), Seasons, and Painters (Anoosheh et al., 2018) dataset in Figure 10. We use the pretrained Stable Diffusion (Rombach et al., 2022) v1-5 checkpoint for the experiment. Note that we bring inpainting masks from the experimental setup of RePaint (Lugmayr et al., 2022). In addition, the masked source image is provided as input, which follows the standard image inpainting setup. As shown, our method achieves superior inpainting performance in diverse scenarios. For the blending operation (denoted * as in Table 1), we follow the setting of BrushNet (Ju et al., 2024), where the binary mask is first blurred using a Gaussian filter before blending.

**Comparison with Stable Diffusion Inpainting.**  We employ Stable Diffusion Inpainting (Rombach et al., 2022) as an additional baseline and compare with ALM. As shown in Table 7, our method consistently outperforms Stable Diffusion Inpainting. Experimental setups are identical as reported in Sec. 4.2.1.

Table 7: Quantitative comparison of our method against Stable Diffusion Inpainting (Rombach et al., 2022) on CelebA-HQ (Karras et al., 2018) and Painters (Anoosheh et al., 2018) dataset.

| Method | Training-free | CelebA-HQ | | | | Painters | | | |
|---|---|---|---|---|---|---|---|---|---|
| | | LPIPS ($\downarrow$) | MSE ($\downarrow$) | SSIM ($\uparrow$) | CS ($\uparrow$) | LPIPS ($\downarrow$) | MSE ($\downarrow$) | SSIM ($\uparrow$) | CS ($\uparrow$) |
| Stable Diffusion Inpainting | N | 0.356 | 0.130 | 0.570 | 26.29 | **0.452** | 0.163 | 0.391 | 26.06 |
| ALM (Ours) | Y | **0.342** | **0.125** | **0.644** | **28.01** | 0.504 | **0.153** | **0.442** | **27.26** |

**Comparison with SDXL-Inpainting.**  We apply ALM to the SDXL, one of the most widely used diffusion models, and compare it with SDXL-Inpainting in Table 8. Our method achieves comparable performance, demonstrating that ALM can immediately upgrade pretrained image generation models to perform high-fidelity versatile content generation purely through our training-free mechanism.

Table 8: Quantitative comparison of our method against SDXL-Inpainting (Podell et al., 2024) on CelebA-HQ (Karras et al., 2018) and Painters (Anoosheh et al., 2018) dataset.

| Method | Training-free | CelebA-HQ | | | | Painters | | | |
|---|---|---|---|---|---|---|---|---|---|
| | | LPIPS ($\downarrow$) | MSE ($\downarrow$) | SSIM ($\uparrow$) | CS ($\uparrow$) | LPIPS ($\downarrow$) | MSE ($\downarrow$) | SSIM ($\uparrow$) | CS ($\uparrow$) |
| SDXL-Inpainting | N | **0.286** | **0.046** | 0.693 | 27.11 | **0.338** | **0.052** | 0.609 | 26.77 |
| ALM (Ours) | Y | 0.315 | 0.105 | **0.710** | **27.78** | 0.422 | 0.133 | **0.618** | **27.50** |

**Failure cases analysis.**  We present failure cases of ALM in Figure 9, where the synthesized unobserved regions are not fully harmonized with the observed content. We hypothesize that this issue arises from the limited capacity of the pretrained Stable Diffusion. To mitigate this limitation,

we fine-tune the model on a specific dataset. Here, considering the high computational cost of full U-Net fine-tuning, we instead adopt LoRA (Hu et al., 2022) with a rank of 4 and attach the resulting weights to the pretrained Stable Diffusion. Interestingly, this lightweight tuning successfully addresses the observed limitations. This demonstrates that our framework is even compatible with and can benefit from existing fine-tuning techniques.

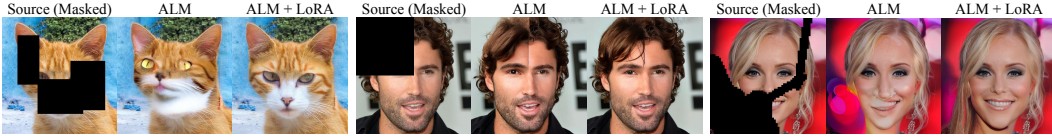

Figure 9: Failure cases of the proposed method. These limitations can be effectively addressed through lightweight per-dataset LoRA (Hu et al., 2022) training.

| Source (Masked) | Inpainted | Source (Masked) | Inpainted | Source (Masked) | Inpainted |
|---|---|---|---|---|---|

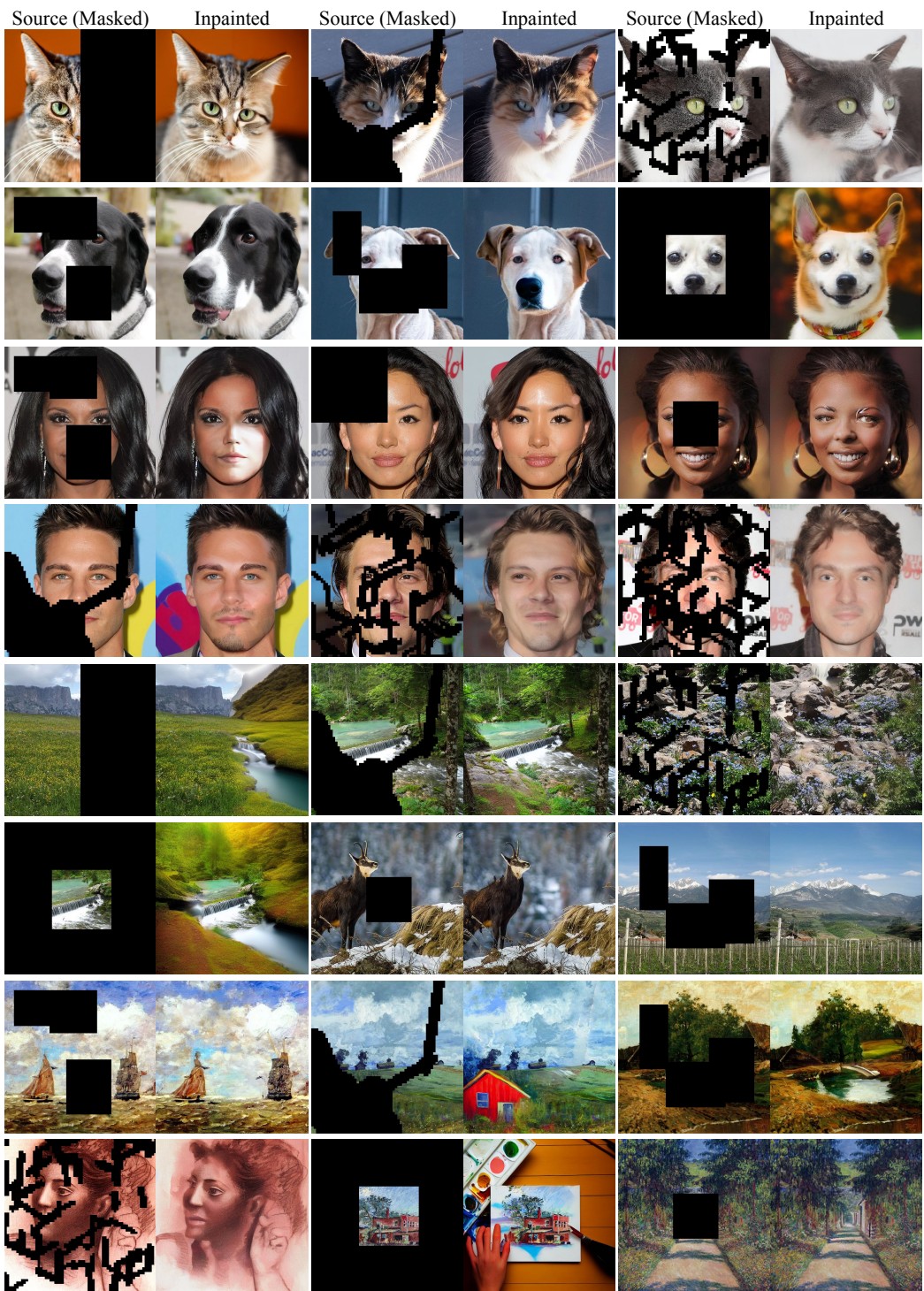

Figure 10: Qualitative results of image inpainting using the pretrained Stable Diffusion (Rombach et al., 2022) on diverse datasets (Choi et al., 2020; Karras et al., 2018; Anoosheh et al., 2018).

## B.2 WIDE IMAGE GENERATION

We used 9 prompts with 50 images per each prompt for evaluation. Figure 11 visualizes diverse samples of wide images generated with ALM using the pretrained Stable Diffusion (Rombach et al., 2022) v2-1-base checkpoint. Our method effectively generates visually plausible images.

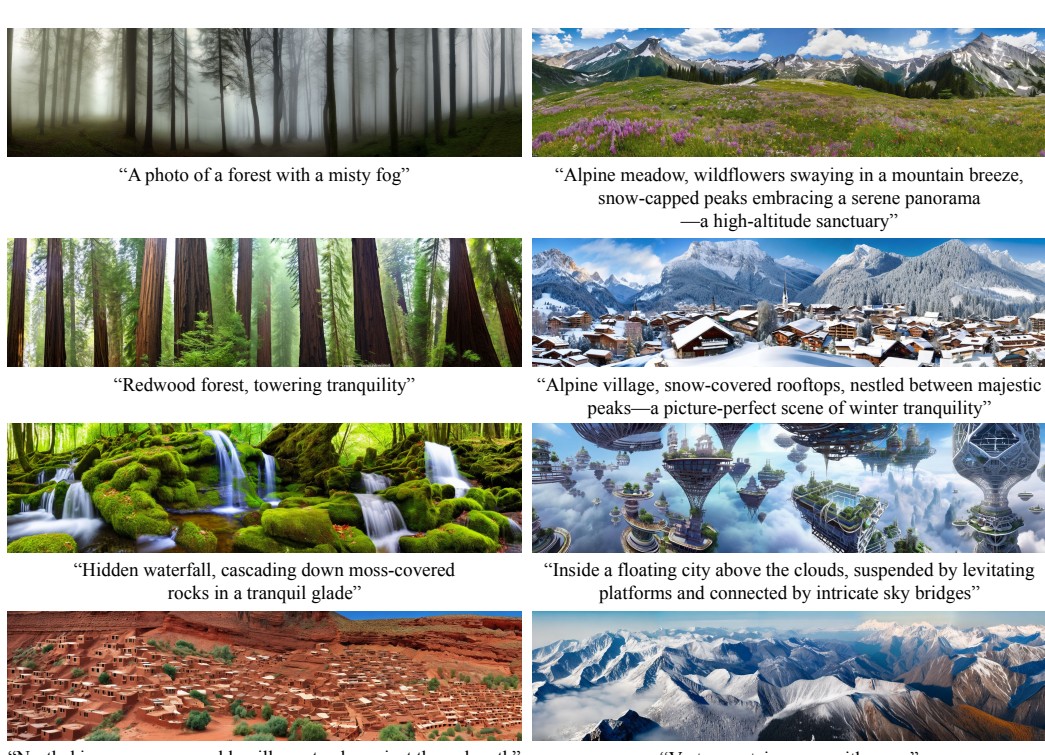

"A photo of a forest with a misty fog"

"Alpine meadow, wildflowers swaying in a mountain breeze, snow-capped peaks embracing a serene panorama —a high-altitude sanctuary"

"Redwood forest, towering tranquility"

"Alpine village, snow-covered rooftops, nestled between majestic peaks—a picture-perfect scene of winter tranquility"

"Hidden waterfall, cascading down moss-covered rocks in a tranquil glade"

"Inside a floating city above the clouds, suspended by levitating platforms and connected by intricate sky bridges"

"Nestled in a canyon, a pueblo village stands against the red earth"

"Vast mountain range with snow"

Figure 11: Qualitative results of wide image generation. We use the pretrained Stable Diffusion (Rombach et al., 2022) to generate $2048 \times 512$ sized wide image.

## B.3 LONG VIDEO GENERATION

Figure 12 shows additional video sequences generated by combining ALM with the pretrained LaVie (Wang et al., 2025) checkpoint. The sequence length is extended from 16 frames to 104 frames. We also provide the video sequences in the submitted Project page.

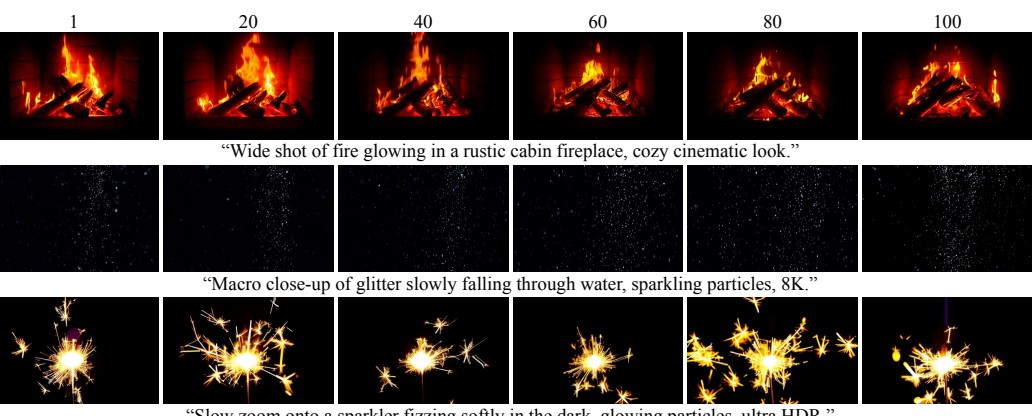

"Wide shot of fire glowing in a rustic cabin fireplace, cozy cinematic look."

"Macro close-up of glitter slowly falling through water, sparkling particles, 8K."

"Slow zoom onto a sparkler fizzing softly in the dark, glowing particles, ultra HDR."

Figure 12: Qualitative results of long video generation. We use the pretrained LaVie (Wang et al., 2025), which by default generates 16-frame videos, and extent them to 104 frames using ALM.

## B.4 HUMAN MOTION INPAINTING

Figure 13 illustrates qualitative results of human motion inpainting under two scenarios: "first-half prediction" (1st–2nd row) and "middle-half prediction" (3rd–4th row), using the pretrained U-Net-based human motion diffusion model (Karunratanakul et al., 2023). ALM effectively reconstructs the unobserved parts of the human motion sequences. We visualize the given frames in orange, and the synthesized frames in blue. For better visualization, we also provide the full motion sequences in the Project page of supplementary material.

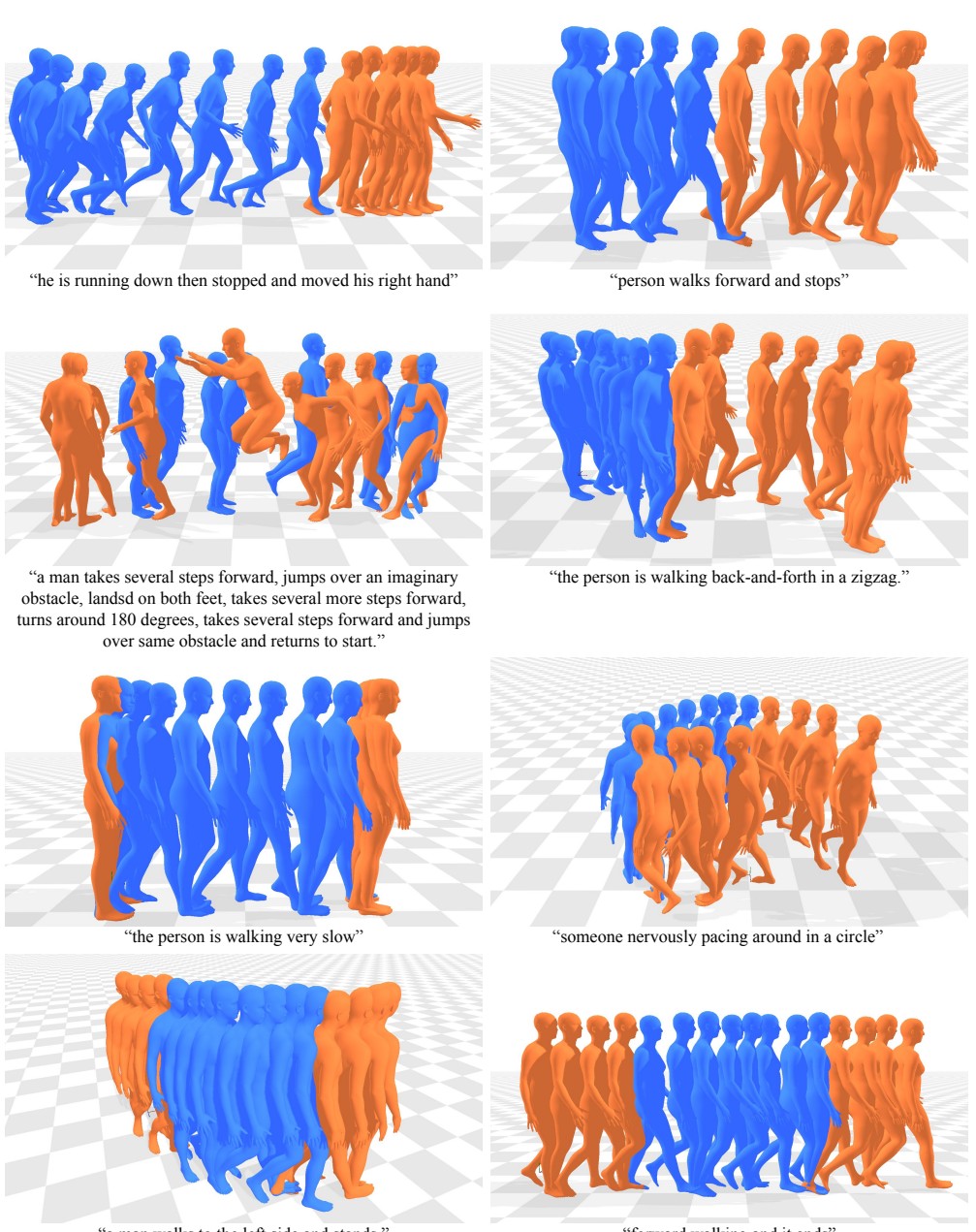

"he is running down then stopped and moved his right hand"

"person walks forward and stops"

"a man takes several steps forward, jumps over an imaginary obstacle, landsd on both feet, takes several more steps forward, turns around 180 degrees, takes several steps forward and jumps over same obstacle and returns to start."

"the person is walking back-and-forth in a zigzag."

"the person is walking very slow"

"someone nervously pacing around in a circle"

"a man walks to the left side and stands."

"forward walking and it ends"

Figure 13: Qualitative results of human motion inpainting. We show first-half inpainting (1st-2nd row) and middle-half completion scenario (3rd-4th row) using the pretrained U-Net-based human motion diffusion model (Karunratanakul et al., 2023).

## B.5 HUMAN MOTION IN-BETWEENING

In this section, we show additional application of ALM. Especially, we evaluate the proposed method on human motion in-betweening task and compare with baselines (Ho et al., 2022; Cohan et al., 2024). In the case of human motion in-betweening, we provide one frame every 10 frames, and the model aims to predict the intermediate sequences. Basically, we follow the experimental setup discussed in Section 4.2.3. We report the measured metrics in Table 9, where our method outperforms Reconstruction Guidance and shows comparable performance with the training-based method, CondMDI.

Table 9: Quantitative comparison on human motion in-betweening task with state-of-the-art baselines Cohan et al. (2024); Ho et al. (2022) using the sequences sampled from HumanML3D (Guo et al., 2022) dataset. Methods with * indicate results obtained with imputation (Tevet et al., 2023).

| Method | Training-free | FID | Matching Score | R-Precision | Diversity | Rank |
|---|---|---|---|---|---|---|
| CondMDI | N | **0.131** | 4.071 | 0.407 | **9.218** | **1** |
| Reconstruction Guidance | Y | 1.965 | 4.293 | 0.372 | 8.105 | 5 |
| Reconstruction Guidance* | Y | 1.703 | 4.270 | 0.376 | 8.217 | 4 |
| Ours | Y | 0.828 | 4.081 | 0.399 | 8.721 | 3 |
| Ours* | Y | 0.943 | **4.034** | **0.411** | 8.679 | 2 |

# C USAGE OF LARGE LANGUAGE MODELS

We used Large Language Models to correct grammatical errors and enhance the overall quality of writing.

# D    ANALYSIS ON HYPERPARAMETER SENSITIVITY

We introduce three hyperparameters: $w_1$, $w_2$ and $w_3$. However, in practice, we fix $w_1 = w_3$ for all experiments and practical usage, yielding only two hyperparameters. We now demonstrate that ALM is robust under variations of these hyperparameters through additional experiments conducted on the image inpainting task.

We sweep $w_1$ over $[0.5, 1, 1.5]$, and $w_2$ over $[0.001, 0.005, 0.01]$ and provide the corresponding quantitative results in Table 10 as well as qualitative comparisons in Figure 14. As shown, our method consistently maintains strong performance across all tested configurations, thereby confirming the robustness of the proposed method.

Table 10: Quantitative analysis of hyperparameter sensitivity on AFHQ (Choi et al., 2020) and CelebA-HQ (Karras et al., 2018) datasets using the pretrained Stable Diffusion (Rombach et al., 2022).

| Method | AFHQ | | | | CelebA-HQ | | | |
|---|---|---|---|---|---|---|---|---|
| | LPIPS (↓) | MSE (↓) | SSIM (↑) | CS (↑) | LPIPS (↓) | MSE (↓) | SSIM (↑) | CS (↑) |
| Baseline (Best) | 0.414 | 0.172 | 0.552 | 28.80 | 0.372 | 0.130 | 0.611 | **28.64** |
| ALM ($w_1$=1.0, $w_2$=0.001) | **0.388** | 0.146 | 0.582 | 28.91 | **0.340** | 0.129 | 0.634 | 28.04 |
| ALM ($w_1$=1.0, $w_2$=0.005) | 0.391 | 0.142 | 0.591 | 29.00 | 0.342 | 0.125 | 0.644 | 28.01 |
| ALM ($w_1$=1.0, $w_2$=0.01) | 0.400 | 0.138 | **0.597** | **29.07** | 0.349 | 0.123 | **0.651** | 27.99 |
| ALM ($w_1$=0.5, $w_2$=0.005) | 0.402 | 0.159 | 0.578 | 28.82 | 0.359 | 0.137 | 0.627 | 28.05 |
| ALM ($w_1$=1.5, $w_2$=0.005) | 0.389 | **0.136** | 0.594 | 29.04 | **0.340** | **0.121** | 0.648 | 28.04 |

| Source (Masked) | (1.0, 0.001) | (1.0, 0.005) | (1.0, 0.01) | (0.5, 0.005) | (1.5, 0.005) |
|---|---|---|---|---|---|

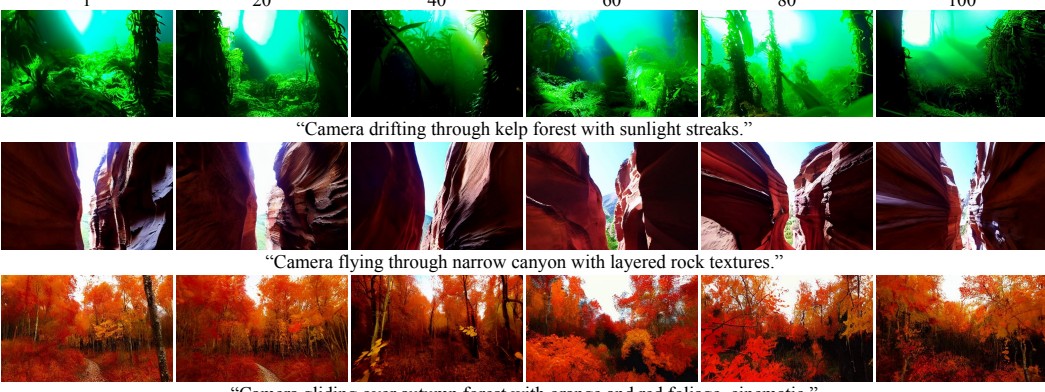

Figure 14: Qualitative result of the proposed method on 5 different hyperparameter values $(w_1, w_2)$. The images are generated using the pretrained Stable Diffusion (Rombach et al., 2022).

# E    APPLICATION OF ALM ON COMPLEX SCENARIOS

In this section, we qualitatively show ALM's performance on challenging scenarios. We first provide the visualizations of long video generation results with dynamic motions in Figure 15, further highlighting the versatility of our method. We further show that our method maintains high performance even when the source prompt is extremely long and packed with semantic details, in Figure 16.

1    20    40    60    80    100

"Camera drifting through kelp forest with sunlight streaks."

"Camera flying through narrow canyon with layered rock textures."

"Camera gliding over autumn forest with orange and red foliage, cinematic."

Figure 15: Qualitative results of ALM on long video generation task with videos containing dynamic motions. We use the pretrained LaVie (Wang et al., 2025) for video generation.

| Source (Masked) | ALM | Source (Masked) | ALM | Source (Masked) | ALM |
|---|---|---|---|---|---|

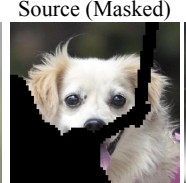 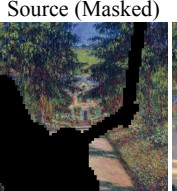 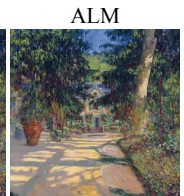

"A photorealistic, high-quality close-up photograph of a cat, captured with a professional DSLR camera in soft natural light. The fur is detailed and textured, with individual hairs visible. Its eyes are sharp, reflective, and expressive, showing clear highlights. Shallow depth of field creates a smooth bokeh background. Warm, balanced colors, realistic shadows."

"A photorealistic, high-quality close-up photograph of a dog, captured with a professional DSLR camera in soft natural light. The fur is detailed and textured, with individual hairs visible. Its eyes are sharp, reflective, and expressive, showing clear highlights. Shallow depth of field creates a smooth bokeh background. Warm, balanced colors, realistic shadows."

"A meticulously captured, high-resolution photograph of a captivating painting. The image showcases the artwork's intricate brushwork, vibrant color palette, and delicate textural details. Every stroke and nuance of the artist's technique is preserved with stunning clarity. The lighting highlights the subtle dimensionality and rich pigments, bringing the painted scene to life through the lens"

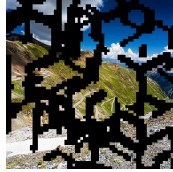 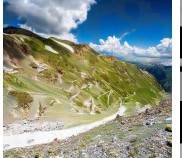 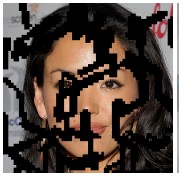 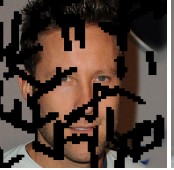 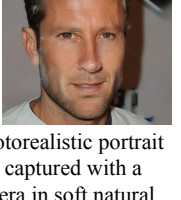

"An epic, sweeping, high-definition photograph of a breathtaking natural landscape. The vast expanse reveals rolling forms and serene features, bathed in soft, ethereal light. Verdant textures carpet the terrain, leading the eye towards distant, majestic horizons under an expansive sky. Every element contributes to a sense of profound tranquility and awe-inspiring grandeur, captured with impeccable clarity."

"A high-quality, photorealistic portrait of a woman face, captured with a professional camera in soft natural light. Her skin texture is detailed and natural, with subtle pores and smooth contours. Her eyes are sharp and expressive with clean reflections. The framing is close-up, using shallow depth of field to create a gentle bokeh background. Balanced colors, soft shadows"

"High-quality, photorealistic portrait of a man's face, captured with a professional camera in soft natural light. Detailed natural skin texture, subtle pores, smooth contours. Sharp, expressive eyes with clean reflections. Close-up framing, shallow depth of field, gentle bokeh background. Balanced colors, soft shadows."

Figure 16: Qualitative result of ALM on complex scenes generated from extremely long and complex source prompts. We use Stable Diffusion XL (Podell et al., 2024) for image generation.

# F Qualitative Ablation on the proposed Acceleration Strategy

We provide qualitative comparisons with and without the acceleration strategy. Notably, the visual quality remains consistent across both setups, demonstrating that the proposed acceleration strategy effectively reduces the computational cost while maintaining the overall performance.

| Source (Masked) | w/o acceleration | ALM (Ours) | Source (Masked) | w/o acceleration | ALM (Ours) |
| --- | --- | --- | --- | --- | --- |

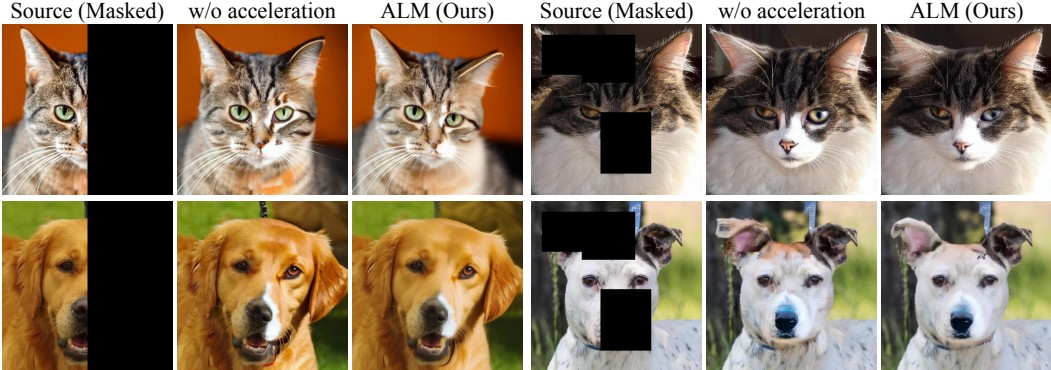

Figure 17: Qualitative comparisons with and without the acceleration strategy. We emphasize that visual quality remains consistent across both setups. This demonstrates that the acceleration strategy effectively reduces the computational cost without performance degradation.

In Table 11, we also show the effectiveness of the proposed acceleration strategy by measuring the required computational cost. As shown, acceleration strategy leads to a significant reduction in runtime.

Table 11: Ablation on acceleration strategy in terms of computational cost.

| Method | GPU Memory (GB) ($\downarrow$) | Runtime (s) ($\downarrow$) |
| --- | --- | --- |
| ALM (Ours) | 4.98 | **8.58** |
| ALM w/o acceleration ($N = 1000$) | 4.98 | 1836.23 |

# G Comparison on Long Video Generation

We compare the proposed method for long video generation against three baselines; FreeNoise (Qiu et al., 2024), SEINE (Chen et al., 2023), and SyncSDE (Lee et al., 2025) using the pretrained LaVie model (Wang et al., 2025). For evaluation, we adopt FVD (Unterthiner et al., 2019), KVD, and CLIP (Radford et al., 2021) text-video similarity. We use a total of 250 video sequences for evaluation, each containing 104 frames. We show the quantitative evaluation result in Table 12. As shown, our method outperforms the baselines, emphasizing the effectiveness of our method. Note that we scale the value of FVD and KVD by 1/1000. We also visualize the qualitative comparison in Figure 18.

Table 12: Quantitative comparison of long video generation with FreeNoise (Qiu et al., 2024), SEINE (Chen et al., 2023), and SyncSDE (Lee et al., 2025) using the pretrained Lavie Wang et al. (2025) model.

| Method | FVD ($\downarrow$) | KVD ($\downarrow$) | CS ($\uparrow$) |
| --- | --- | --- | --- |
| FreeNoise | 2.552 | 4.360 | 31.08 |
| SEINE | 3.650 | 6.611 | 30.43 |
| SyncSDE | 2.505 | 5.635 | **31.18** |
| ALM (Ours) | **2.487** | **4.219** | 31.12 |

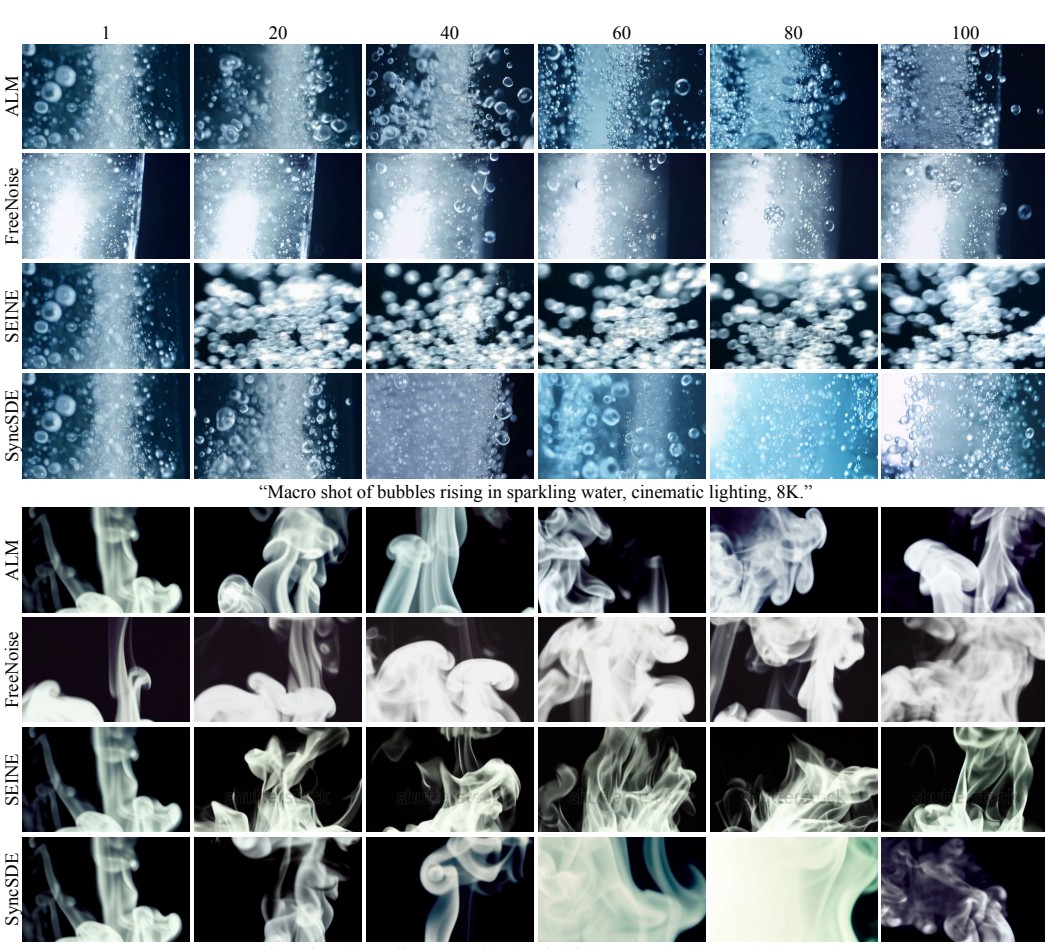

Figure 18: Qualitative comparisons in the task of long video generation. We compare our method with FreeNoise (Qiu et al., 2024), SEINE (Chen et al., 2023), and SyncSDE (Lee et al., 2025). Our method shows superior performance compared to the baselines, while baselines struggle to generate temporally consistent outcomes.

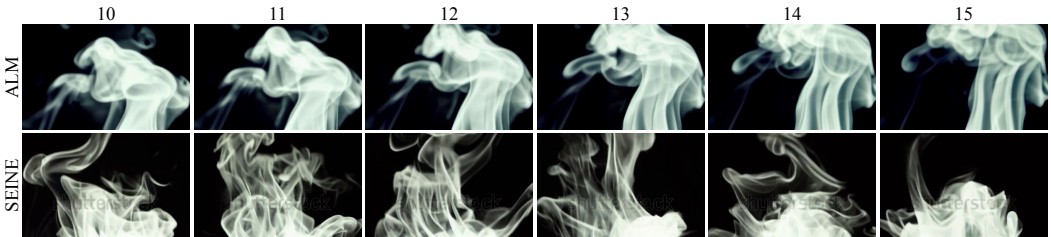

Figure 19: Frame-by-frame visualization of the long video generated by ALM and SEINE (Chen et al., 2023). Our method well preserves the temporal identity and evolution of the smoke structure, while SEINE induces abnormal structural reformation between adjacent frames.

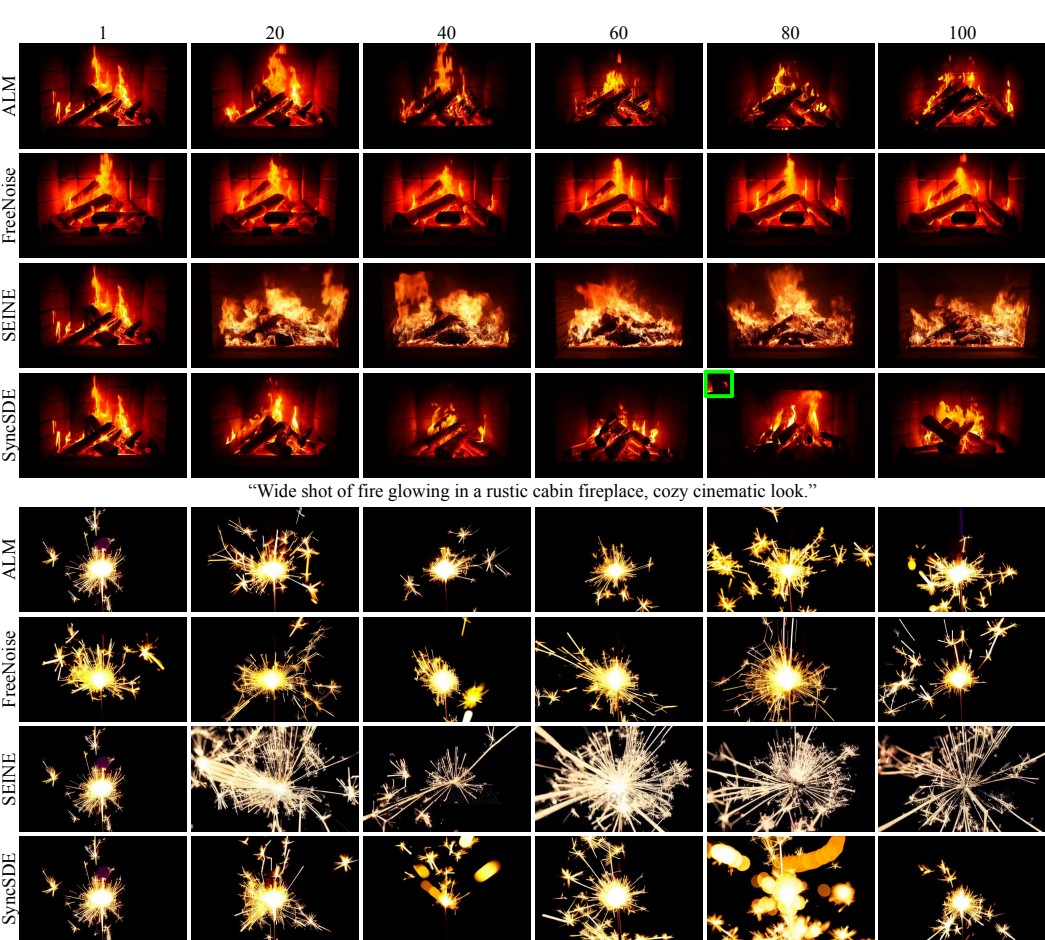

Figure 20: Additional qualitative comparisons in the task of long video generation. We compare our method with FreeNoise (Qiu et al., 2024), SEINE (Chen et al., 2023), and SyncSDE (Lee et al., 2025). Our method shows superior performance compared to the baselines, while baselines struggle to generate temporally consistent outcomes.

