# OpenReview forum: "Accelerated Likelihood Maximization for Diffusion-based Versatile Content Generation"
_ICLR.cc/2026/Conference — Submitted to ICLR 2026_

### Official Review · Reviewer_bVRd · 2025-10-29

**Soundness:** 2
**Presentation:** 2
**Contribution:** 2
**Rating:** 6
**Confidence:** 4

**Summary:**

The paper introduces Accelerated Likelihood Maximization (ALM), a novel, training-free approach designed to enhance the quality and coherence of content generated by pre-trained diffusion models when conditioned on partial inputs (e.g., masked images, keyframes).

The central problem ALM addresses is the trade-off between existing methods: training-based approaches are costly and generalize poorly, while existing training-free methods often struggle to maintain global consistency, enforcing constraints only locally. ALM solves this by formulating the conditional generation as an efficient optimization problem in the latent space. It aims to maximize the likelihood of the generated sample given the observed constraints, using an accelerated optimization scheme to converge quickly.

The authors demonstrate ALM's versatility across multiple domains, including image inpainting/outpainting, 3D object completion, and challenging temporal tasks like human motion in-betweening. Quantitatively, the method shows significant improvements over traditional training-free baselines like Reconstruction Guidance, achieving performance comparable to training-based methods on several metrics.

**Strengths:**

1. Effective Training-Free Paradigm: The core contribution, Accelerated Likelihood Maximization (ALM), is a highly effective, non-intrusive method. It successfully leverages the power of existing, large-scale pre-trained diffusion models without requiring task-specific fine-tuning, which significantly reduces computational cost and improves accessibility.

2. Superior Coherence and Consistency: ALM is shown to produce samples with better global consistency than other training-free conditional methods (e.g., standard Reconstruction Guidance). From the quantitative results from Table 1, it can also beat training-based methods.

3. Demonstrated Versatility: The paper provides compelling evidence of ALM's broad applicability. Its success across diverse modalities—from static 2D images to 3D point clouds and dynamic human motion sequences—suggests a highly robust and generalizable algorithm, which is a key advantage over specialized models.

**Weaknesses:**

1. Analysis of Computational Overhead: While the method is termed "Accelerated," the primary weakness is the lack of a clear, comparative analysis of the inference wall-clock time. Since ALM involves an iterative optimization process during inference (unlike single-pass guidance or simple sampling), the paper must provide stronger evidence showing the method’s actual speed and efficiency relative to competing constrained generation techniques.

2. Sensitivity to Optimization Hyperparameters: Optimization-based sampling methods can be highly sensitive to tuning parameters (e.g., $w_1,w_2, w_3$ in Eq (1)). The paper would be strengthened by including a dedicated ablation study detailing the sensitivity of results to these key optimization parameters, which is crucial for reproducibility and practical deployment.

**Questions:**

1. See weaknesses for the questions of computational overhead and sensitivity of hyper parameters.

2. The main question is the significance of this work. **Currently, visual foundation models, such as GPT-Image, Nano-banana, Seedream, Qwen-Image, and Hunyuan-Image, can already conduct versatile content generation, including in-painting, out-painting, etc.** While the quantitative results demonstrate ALM's superiority over existing training-free and task-specific training-based baselines, the paper lacks a direct comparison against state-of-the-art large foundational models (e.g., the inpainting/outpainting capabilities of Qwen-Image). **Given that the research and industrial communities are increasingly adopting these high-performing foundational models, can the authors better contextualize the practical significance of ALM?** Specifically, how does ALM's output quality and efficiency practically compare to the best available models in scenarios like large-mask inpainting, even those that are closed-source?

---

> ### Author Response · Authors · 2025-11-22
> **Response to Reviewer bVRd (1/2)**
>
> We thank you for spending valuable time reviewing our paper.
>
> **W1.** *Analysis of Computational Overhead.*
>
> **A1.** In the answer #3 of the global response, we analyze the computational cost of ALM and the baseline methods. Table D1 reports the required GPU memory (GB) and runtime (s) for generating a single image. Overall, **our method exhibits computational overhead comparable to existing baselines.** We note that although ALM incurs a slightly longer runtime than SyncSDE due to the additional sampling of $\epsilon_{\theta}(\mathbf{E}_t, t)$, the difference is minor and ALM achieves better performance than SyncSDE [D1]. With the results in Table 1 of the main paper, these findings highlight that **ALM achieves the best trade-off between performance and computational efficiency** among the various methods.
>
> Table D1. Computational cost analysis.
>
> | Method | Additional training cost  | [Sampling] GPU Memory (GB) | [Sampling] Runtime (s/image) | LPIPS (↓) |
> |--|--|--|--|--|
> | BrushNet  [G8]  | 3 days with 8 V100 GPUs  | 4.73   | 3.189  | 0.434 |
> | PowerPaint  [G9] | requires 8 A100 GPUs | 5.54  | 4.089  | 0.428 |
> | SyncSDE  [G6]  | - | 4.98   | 6.734   | 0.414 |
> | BLD   [G7]  | -  | 3.39 | 2.615    | 0.421 |
> | HD-Painter [G10]    | - | 29.07   | 38.840   | 0.398 |
> | ALM (Ours)    | - | 4.98 | 8.584   | 0.391 |
> | ALM w/o inversion| -   | 4.98   | 3.850   | - |
>
>
> **W2.** *Sensitivity to Optimization Hyperparameters.*
>
> **A2.** In the answer #2 of the global response, we analyze the effects of each hyperparameter and show that our method is robust to the change of hyperparameters. Specifically, we sweep $w_1$ over $[0.5, 1, 1.5]$, and $w_2$ over $[0.001, 0.005, 0.01]$ and provide the corresponding quantitative results in Table D2-D3 as well as qualitative comparisons in the Section D of the revised PDF. Note that the default setting used in the main paper is $w_1 = 1$ and $w_2 = 0.005$. As shown, our method consistently maintains strong performance across all tested configurations, still outperforming the baseline methods, thereby confirming the robustness of the proposed method.
>
> Table D2. Quantitative analysis of the hyperparameter sensitivity on AFHQ dataset [D6].
>
> | Method                          | LPIPS ↓ | MSE ↓  | SSIM ↑ | CS ↑  |
> |----------------------------------|---------|--------|--------|-------|
> | Baseline (SyncSDE [D1], Best)                  |  0.414   |  0.172  | 0.552  |  28.80 |
> | ALM ($w_1=1.0$, $w_2=0.001$)           | **0.388** | 0.146  | 0.582  | 28.91 |
> | ALM ($w_1=1.0$, $w_2=0.005$)           | 0.391   | 0.142  | 0.591  | 29.00 |
> | ALM ($w_1=1.0$, $w_2=0.01$ )           | 0.400   | 0.138  | **0.597** | **29.07** |
> | ALM ($w_1=0.5$, $w_2=0.005$)           | 0.402   | 0.159  | 0.578  | 28.82 |
> | ALM ($w_1=1.5$, $w_2=0.005$ )          | 0.389   | **0.136** | 0.594  | 29.04 |
>
>
> Table D3. Quantitative analysis of the hyperparameter sensitivity on CelebA-HQ dataset [D7].
>
> | Method                          | LPIPS ↓ | MSE ↓  | SSIM ↑ | CS ↑  |
> |----------------------------------|---------|--------|--------|--------|
> | Baseline (BLD [D4], Best)                           | 0.372   | 0.130  | 0.611  | **28.64** |
> | ALM ($w_1=1.0$, $w_2=0.001$ )          | **0.340** | 0.129  | 0.634  | 28.04 |
> | ALM ($w_1=1.0$, $w_2=0.005$  )         | 0.342   | 0.125  | 0.644  | 28.01 |
> | ALM ($w_1=1.0$, $w_2=0.01$  )          | 0.349   | 0.123  | **0.651** | 27.99 |
> | ALM ($w_1=0.5$, $w_2=0.005$ )          | 0.359   | 0.137  | 0.627  | 28.05 |
> | ALM ($w_1=1.5$, $w_2=0.005$ )          | **0.340** | **0.121** | 0.648  | 28.04 |

---

> ### Author Response · Authors · 2025-11-22
> **Response to Reviewer bVRd (2/2)**
>
> **Q1.** *The main question is the significance of this work. Given that the research and industrial communities are increasingly adopting these high-performing foundational models, can the authors better contextualize the practical significance of ALM?*
>
> **A3.** We thank you for raising this valuable point. As discussed in the answer #1 of the global response, recent visual foundation models indeed support versatile content generation such as inpainting and outpainting. However, achieving such capabilities typically requires substantial task-specific training with large datasets and heavy computational resources.
>
> In contrast, ALM offers a fundamentally different value proposition. **Our method is fully training-free and operates entirely on top of any pretrained foundation model, effectively extending the applicability of modern foundation models beyond generation.** For example, in the case of the image domain, ALM transforms powerful generative models (*e.g.*, SD [D8], SDXL [D9]) into highly capable content-editing models without any additional training, datasets, or proprietary tooling. This completely eliminates the heavy computational and engineering overhead required by existing approaches.
>
> To demonstrate this, we evaluate ALM on Stable Diffusion XL and compare it with SD-Inpainting and SDXL-Inpainting (Tables D4–D7). ALM consistently outperforms SD-Inpainting and performs comparably to SDXL-Inpainting, **showing that our training-free mechanism effectively upgrades pretrained generators into versatile content-editing models.**
>
> In summary, while foundation models may support limited editing, ALM uniquely provides a practical, widely applicable, and training-free approach for high-quality content manipulation.
>
> Table D4. Quantitative comparison of the proposed method with SD-Inpainting on CelebA-HQ dataset [D7].
>
> | Method  | Training-free | LPIPS ↓ | MSE ↓  | SSIM ↑ | CS ↑   |
> |---|---|---|--|--|--|
> | SD-Inpainting | N  | 0.356   | 0.130  | 0.570  | 26.29  |
> | ALM (Ours)    | Y   | **0.342**   | **0.125**  | **0.644**  | **28.01**  |
>
>
> Table D5. Quantitative comparison of the proposed method with SD-Inpainting on Painters dataset [D10].
>
> | Method   | Training-free | LPIPS ↓ | MSE ↓  | SSIM ↑ | CS ↑   |
> |---|----|--|---|--|--|
> | SD-Inpainting | N    | **0.452**   | 0.163  | 0.391  | 26.06 |
> | ALM (Ours)    | Y   | 0.504   | **0.153**  |**0.442**  | **27.26** |
>
> Table D6. Quantitative comparison of the proposed method with SDXL-Inpainting on CelebA-HQ dataset [D7].
>
> | Method | Training-free | LPIPS ↓ | MSE ↓  | SSIM ↑ | CS ↑   |
> |---|---|-|--|--|-|
> | SDXL-Inpainting  | N   | **0.286**  | **0.046**  | 0.693  | 27.11 |
> | ALM (Ours) | Y  | 0.315   | 0.105  | **0.710**  |**27.78** |
>
> Table D7. Quantitative comparison of the proposed method with SDXL-Inpainting on Painters dataset [D10].
>
> | Method           | Training-free | LPIPS ↓ | MSE ↓  | SSIM ↑ | CS ↑   |
> |---|--|-|-|--|--|
> | SDXL-Inpainting  | N  | **0.338**   |**0.052**  | 0.609  | 26.77 |
> | ALM (Ours)   | Y   | 0.422   | 0.133  | **0.618** |**27.50** |
>
> In summary, while foundation models can sometimes perform specific editing tasks, ALM uniquely provides a training-free, and widely applicable mechanism that turns any pretrained generator into a flexible content manipulation model, without additional computation.
>
>
> We hope our explanations have clarified the issues, and we welcome any further questions.
>
>
> **References**
>
> [D1] Lee et al., SyncSDE: A probabilistic framework for diffusion synchronization. CVPR, 2025.
>
> [D2] Ju et al., BrushNet: A plug-and-play image inpainting model with decomposed dual-branch diffusion. ECCV, 2024.
>
> [D3] Zhuang et al., A task is worth one word: Learning with task prompts for high-quality versatile image inpainting. ECCV, 2024.
>
> [D4] Avrahami et al., Blended latent diffusion. ACM TOG, 2023.
>
> [D5] Manukyan et al., HD-Painter: High-resolution and prompt-faithful text-guided image inpainting with diffusion models. ICLR, 2025.
>
> [D6] Choi et al., StarGAN v2: Diverse image synthesis for multiple domains. CVPR, 2020.
>
> [D7] Karras et al., Progressive growing of GANs for improved quality, stability, and variation. ICLR, 2018.
>
> [D8] Rombach et al., High-resolution image synthesis with latent diffusion models. CVPR, 2022.
>
> [D9] Podell et al., SDXL: Improving latent diffusion models for high-resolution image synthesis. ICLR, 2024.
>
> [D10] Anoosheh et al., ComboGAN: Unrestrained scalability for image domain translation. CVPRW, 2018.

---

### Official Review · Reviewer_bmgX · 2025-10-30

**Soundness:** 2
**Presentation:** 3
**Contribution:** 2
**Rating:** 4
**Confidence:** 4

**Summary:**

The paper proposes Accelerated Likelihood Maximization (ALM), a training-free sampling strategy for diffusion models that treats the masked (unobserved) region as an explicit variable to be optimized during reverse diffusion. Concretely, the update augments DDIM with three terms: preservation of observed content, and two guidance terms derived from maximizing a conditional and a joint likelihood, respectively. A one-step approximation replaces an inner iterative optimization by assuming small updates and Lipschitzness of the noise predictor. The method is demonstrated on image inpainting/outpainting, wide-image generation, long-video generation, and 3D human-motion inpainting.

**Strengths:**

ALM is simple to implement on top of pretrained diffusion models, requires no additional training in the common case, and unifies a set of conditional generation tasks under one sampling rule. The separation into conditional vs. joint likelihood terms is well-motivated and supported by ablations, and the one-step approximation offers a practical speedup over inner loops conceptually required by likelihood maximization.

**Weaknesses:**

*Core reliance on strong approximations; limited guarantees.*
The key one-step acceleration hinges on two claims rather than verifiable conditions: sufficiently small update magnitude and Lipschitz noise predictor; the resulting argument shows only $O(\Delta Y_t)$ closeness between consecutive inner steps, not a convergence guarantee for the surrogate optimization. This weakens theoretical backing for using a single step across diverse regimes.

*Heuristic factorization of the joint term.*
In the derivation, the joint likelihood gradient is computed by assuming $p(X_t, M, Y_t \mid c) \approx p(E_t \mid c) \quad \text{with} \quad \mathbf{E}_t = X_t + Y_t \odot M$ ,which is convenient but potentially biased when the blending operation misaligns with the pretrained model’s data manifold or when mask boundaries impose nonlocal correlations. The paper does not quantify the error of this approximation.

*Hyperparameter sensitivity without principled schedules.*
The update uses three weights $\omega_1,\omega_2,\omega_3$ (with $\omega_2=N\lambda_1$, $\omega_3=N(\lambda_1-\lambda_2)$)
and an empirical decaying schedule with $\sigma_t$; no guidance is given for choosing or adapting these beyond heuristics. This raises concerns about robustness across models, datasets, and mask shapes.

*“Training-free” claim is brittle in practice.*
The paper itself shows failure cases where the synthesized unobserved regions are not harmonized with observed content; resolving them required LoRA fine-tuning of the base model. This suggests that success depends on base-model coverage and that ALM can need per-domain tuning to reach advertised quality.

*Long-video results are qualitative only.*
The paper showcases visual examples for extending short clips to 104 frames but provides no quantitative metrics (e.g., FVD, temporal consistency) or runtime, leaving claims of temporal coherence and efficiency insufficiently substantiated. Including the evaluation on VBench is welcome.

**Questions:**

Can you provide measurable criteria (in terms of mask size, noise level, or gradient norms) under which the one-step surrogate is provably faithful to the inner loop? Any failure rates when the small-update assumption is violated?

Can you quantify or bound the error incurred by this approximation, or at least report an ablation where this factorization is replaced by a learned or Monte-Carlo estimate to assess sensitivity?

do you have a principled schedule or adaptive rule (e.g., based on validation or on-the-fly score magnitudes)? How sensitive are results to $\omega_1, \omega_2, \omega_3$ ratios across datasets and mask geometries?

In the failure cases that required LoRA, how often does this occur in practice, and what performance delta does LoRA provide relative to pure ALM? Please clarify the boundary of “training-free” claims with quantitative evidence.

Please report end-to-end runtime (with/without inversion), and a study of inversion error vs. observed-region PSNR/SSIM to demonstrate that preservation remains reliable under realistic noise/model choices.

For long-video generation, can you add FVD/Fréchet Video Distance and temporal consistency metrics, plus throughput (FPS) on standardized hardware, to substantiate the efficiency and coherence claims?

---

> ### Author Response · Authors · 2025-11-22
> **Response to Reviewer bmgX (1/5)**
>
> **W1.** *Core reliance on strong approximations; limited guarantees.*
>
> **A1.** We appreciate the reviewer’s observation regarding the reliance on the small-update assumption and the Lipschitz continuity of the noise prediction network. We agree that these conditions are assumptions rather than quantities that can be directly verified for an arbitrary pretrained model. Nonetheless, our formulation is closely aligned with prior theoretical justifications of perturbation stability in diffusion models, most notably FIFO-Diffusion [C1] where similar boundedness and Lipschitz assumptions are adopted to justify step-wise consistency of the denoising dynamics in video generation.
>
> While we also acknowledge that this analysis does not constitute a full convergence guarantee of the surrogate optimization, **it provides a theoretically meaningful justification that the inner-loop updates remain within an $O(\Delta Y_t)$ neighborhood across iterations, precisely the setting in which our one-step approximation is valid.** Importantly, this small-update regime aligns well with empirical behavior observed in the pretrained diffusion models, as our method consistently achieves high performance even when using the proposed one-step approximation strategy.
>
>
> **W2.** *Heuristic factorization of the joint term.*
>
> **A2.** We sincerely thank the reviewer’s insightful comment regarding the heuristic factorization. Our formulation $p(X_t, M, Y_t \mid c) \approx p(E_t \mid c)$ is indeed an approximation introduced to make the optimization tractable, but it is not arbitrary. The key design principle of our method is to operate strictly within a *small-update*, ensuring that the blended variable $E_t = X_t + Y_t \odot M$ remains close to the pretrained diffusion model’s data manifold. In this scheme, the model exhibits locally smooth behavior, and the factorization provides an accurate approximation of $p(X_t, M, Y_t \mid c)$.
>
> While we acknowledge that this approximation is challenging to quantify analytically, our empirical results consistently demonstrate that the assumed factorization does not lead to degradation. In fact, across all experimental settings, the proposed method produces high-quality results without introducing significant artifacts. This strongly suggests that the adopted approximation is not only reasonable but also effectively optimal for the practical scenarios in which ALM is applied.

---

> ### Author Response · Authors · 2025-11-22
> **Response to Reviewer bmgX (2/5)**
>
> **W3.** *Hyperparameter sensitivity without principled schedules.*
>
> **A3.** In the answer #2 of the global response, we first analyze the effects of each hyperparameter and show that our method is robust to the change of hyperparameters. Specifically, we sweep $w_1$ over $[0.5, 1, 1.5]$, and $w_2$ over $[0.001, 0.005, 0.01]$ and provide the corresponding quantitative results in Table C1-C2 as well as qualitative comparisons in the Section D of the revised PDF. Note that the default setting used in the main paper is $w_1 = 1$ and $w_2 = 0.005$. As shown, our method consistently maintains strong performance across all tested configurations, still outperforming the baseline methods, thereby confirming the robustness of the proposed method.
>
> Table C1. Quantitative analysis of the hyperparameter sensitivity on AFHQ dataset [C2].
>
> | Method| LPIPS ↓ | MSE ↓  | SSIM ↑ | CS ↑  |
> |-|-|-|-|-|
> | Baseline (SyncSDE [C3], Best)  |  0.414  |  0.172  | 0.552  |  28.80 |
> | ALM ($w_1=1.0$, $w_2=0.001$)  | **0.388** | 0.146  | 0.582  | 28.91 |
> | ALM ($w_1=1.0$, $w_2=0.005$) | 0.391 | 0.142 | 0.591  | 29.00 |
> | ALM ($w_1=1.0$, $w_2=0.01$ ) | 0.400  | 0.138 | **0.597** | **29.07** |
> | ALM ($w_1=0.5$, $w_2=0.005$) | 0.402  | 0.159 | 0.578  | 28.82 |
> | ALM ($w_1=1.5$, $w_2=0.005$ )  | 0.389 | **0.136** | 0.594  | 29.04 |
>
>
> Table C2. Quantitative analysis of the hyperparameter sensitivity on CelebA-HQ dataset [C4].
>
> | Method   | LPIPS ↓ | MSE ↓  | SSIM ↑ | CS ↑  |
> |-|-|-|-|-|
> | Baseline (BLD [C5], Best) | 0.372 | 0.130 | 0.611 | **28.64** |
> | ALM ($w_1=1.0$, $w_2=0.001$) | **0.340** | 0.129  | 0.634  | 28.04 |
> | ALM ($w_1=1.0$, $w_2=0.005$) | 0.342  | 0.125  | 0.644  | 28.01 |
> | ALM ($w_1=1.0$, $w_2=0.01$) | 0.349 | 0.123 | **0.651** | 27.99 |
> | ALM ($w_1=0.5$, $w_2=0.005$)  | 0.359 | 0.137  | 0.627  | 28.05 |
> | ALM ($w_1=1.5$, $w_2=0.005$)  | **0.340** | **0.121** | 0.648  | 28.04 |
>
>
> Secondly, We further demonstrate that **ALM’s performance does not rely on the empirical $\sigma_t$-based decay schedule**. To analyze this, we compare the results with and without the $\sigma_t$-based decay in Table C3-C4. As shown, the two setups exhibit nearly identical performance, confirming that the empirical $\sigma_t$ scheduling is **not** a critical component of our method’s effectiveness. Here, we note that we empirically employ a decaying schedule to better ensure that the Claim 1 (small-update assumption) of Section 3.4. Remains valid throughout sampling.
>
> Table C3. Quantitative analysis of the effect of $\sigma_t$-based decay on AFHQ dataset [C2].
>
> | Method  | LPIPS ↓ | MSE ↓  | SSIM ↑ | CS ↑  |
> |-|-|-|-|-|
> | ALM  | 0.391 | 0.142 | 0.591  | 29.00 |
> | ALM w/o decay | 0.392 | 0.125  | 0.607 | 29.25 |
>
> Table C4. Quantitative analysis of the effect of $\sigma_t$-based decay on CelebA-HQ dataset [C4].
>
> | Method  | LPIPS ↓ | MSE ↓  | SSIM ↑ | CS ↑  |
> |-|-|-|-|-|
> | ALM  | 0.342 | 0.125 | 0.644  | 28.01 |
> | ALM w/o decay | 0.346  | 0.114  | 0.660  | 28.35 |
>
> Thirdly, we show that our method is robust across model (backbones) by applying our method on both SD [C6] and SDXL [C7], as introduced in the answer #1 of the global response. We also quantitatively compare ALM against SD-Inpainting (Table C5-C6), and SDXL-Inpainting (Table C7-C8) using the same setup described in the main paper. Our method outperforms SD-Inpainting, and achieves comparable performance compared to SDXL-Inpainting.
>
> Table C5. Quantitative comparison of the proposed method with SD-Inpainting on CelebA-HQ dataset [C4].
>
> | Method   | Training-free | LPIPS ↓ | MSE ↓  | SSIM ↑ | CS ↑   |
> |--|--|--|--|--|--|
> | SD-Inpainting | N | 0.356 | 0.130 | 0.570 | 26.29 |
> | ALM (Ours)  | Y  | **0.342** | **0.125** | **0.644**  | **28.01**  |
>
> Table C6. Quantitative comparison of the proposed method with SD-Inpainting on Painters dataset [C8].
>
> | Method   | Training-free | LPIPS ↓ | MSE ↓  | SSIM ↑ | CS ↑   |
> |--|--|--|-|--|--|
> | SD-Inpainting | N | **0.452** | 0.163  | 0.391 | 26.06 |
> | ALM (Ours)  | Y | 0.504 | **0.153* |**0.442** | **27.26** |
>
> Table C7. Quantitative comparison of the proposed method with SDXL-Inpainting on CelebA-HQ dataset [C4].
>
> | Method    | Training-free | LPIPS ↓ | MSE ↓  | SSIM ↑ | CS ↑   |
> |--|--|--|--|--|--|
> | SDXL-Inpainting  | N | **0.286** | **0.046** | 0.693  | 27.11 |
> | ALM (Ours)  | Y | 0.315   | 0.105 | **0.710** |**27.78** |
>
> Table C8. Quantitative comparison of the proposed method with SDXL-Inpainting on Painters dataset [C8].
>
> | Method   | Training-free | LPIPS ↓ | MSE ↓  | SSIM ↑ | CS ↑   |
> |--|--|--|--|--|--|
> | SDXL-Inpainting  | N | **0.338** |**0.052**  | 0.609  | 26.77 |
> | ALM (Ours)  | Y| 0.422   | 0.133  | **0.618** |**27.50** |
>
> Lastly, we note that we evaluated the proposed method on 4 different datasets and 10 distinct mask shapes for image inpainting (and even more diverse cases when including wide image generation and human motion completion). These experiments demonstrate the robustness of our method across both datasets and mask shapes.

---

> ### Author Response · Authors · 2025-11-22
> **Response to Reviewer bmgX (3/5)**
>
> **W4.** *“Training-free” claim is brittle in practice.*
>
> **A4.** Our method is training-free in the sense that it requires no additional training or optimization. As demonstrated throughout the main paper, ALM operates effectively without any LoRA [C9] weight training. The LoRA-based results were included in the Appendix **not to imply that ALM depends on training, but rather to show that our framework is compatible with and can benefit from existing fine-tuning techniques.** In other words, LoRA is optional and serves only as an additional enhancement pathway of our method.
>
>
> **W5.** *Long-video results are qualitative only.*
>
> **A5.** In the answer #6 of the global response, we compare the proposed method for long video generation task against three baselines; FreeNoise [C10], SEINE [C11], and SyncSDE [C3] using the pretrained LaVie model [C12]. For evaluation, we adopt FVD [C13], KVD, and CLIP [C14] text-video similarity. We use a total of 250 video sequences for evaluation, each containing 104 frames. We show the quantitative evaluation result in Table C9. As shown, our method outperforms the baselines, emphasizing the effectiveness of our method. Note that we scale the value of FVD and KVD by 1/1000. We also visualize the qualitative comparison in the Section G of the revised PDF file.
>
> Table C9. Quantitative evaluation of long video generation
>
> | Method   | FVD ↓ | KVD ↓  | CS ↑  |
> |---|---|--|----|
> | FreeNoise [C10]  | 2.5516   | 4.3600   | 31.08   |
> | SEINE   [C11]  | 3.6496   | 6.6111   | 30.43   |
> | SyncSDE    [C3]  | 2.5049   | 5.6352   | **31.18**   |
> | ALM (Ours)  | **2.4870** | **4.2188** | 31.12 |
>
>
> **Q1.** *Can you provide measurable criteria (in terms of mask size, noise level, or gradient norms) under which the one-step surrogate is provably faithful to the inner loop? Any failure rates when the small-update assumption is violated?*
>
> **A6.** In our experimental setup, the one-step surrogate remains accurate even when the average norm of $\Delta Y_t$ is as small as $0.00815$ (which corresponds to the $N=1000$ case), **which indicates that the approximation holds well in practical settings.** As discussed in Section 3.4. of the main paper, our acceleration strategy is derived under the small-update assumption $|\Delta Y_t^i| << 1$ and the Lipschitz continuity of the pretrained noise prediction network. These conditions ensure that the sequence $\{ \Delta Y_t^i \}_{i=1}^N$ remains approximately constant and can therefore be collapsed into a single step update (Eq. (24)). However, when these assumptions are violated, for example, by choosing excessively large $w_1$ and $w_2, the one-step approximation would no longer hold. In such cases, our method may produce unrealistic samples.
>
>
> **Q2.** *Can you quantify or bound the error incurred by this approximation, or at least report an ablation where this factorization is replaced by a learned or Monte-Carlo estimate to assess sensitivity?*
>
> **A7.** In practice, it’s challenging to directly quantify the approximation error using either a learned estimator or Monte Carlo sampling. Training such an estimator would require (pseudo-)ground truth supervision, which is not available for our setting. We acknowledge this limitation. However, we emphasize that even with this approximation, our method consistently performs well across all evaluated four different datasets and mask shapes. This strong empirical robustness suggests that the error introduced by the factorization is small in practice, and does not meaningfully affect the quality of the resulting samples.
>
>
> **Q3.** *Do you have a principled schedule or adaptive rule (e.g., based on validation or on-the-fly score magnitudes)? How sensitive are results to $w_1$, $w_2$, and $w_3$ ratios across datasets and mask geometries?*
>
> **A8.** In our work, we adopt a simple $\sigma_t$-based decaying, to help maintain the small-update assumption (Claim 1 in Section 3.4. of the main paper), as also discussed in Answer #3.  Importantly, we show in A3 that our method is robust to the specific hyperparameter values or scheduling strategy, and it consistently shows superior performance across a wide range of experimental settings. This suggests that a more principled or adaptive schedule is not required for the method to perform well in practice.
>
> Furthermore, we tested our method on 10 different mask shapes for image inpainting (and even more when including wide image generation and human motion completion). Across all these cases, our method consistently outperforms the baselines, indicating that performance is not sensitive to $w_1$, $w_2$, $w_3$, and mask geometries.

---

> ### Author Response · Authors · 2025-11-22
> **Response to Reviewer bmgX (4/5)**
>
> **Q4.** *In the failure cases that require LoRA, how often does this occur in practice, and what performance delta does LoRA provide relative to pure ALM? Please clarify the boundary of “training-free” claims with quantitative evidence.*
>
> **A9.** We empirically observe that failure cases become more likely when the mask covers a very large portion of the image. However, even in such cases, the failure frequency remains low. For instance, in a wide image generation task, we use a stride of 48 for a latent width of 64, meaning that the masked region covers 3/4 of the image area. Despite this large mask, our method still produces high-quality results and strong quantitative performance on wide image generation (as demonstrated in the Section 4), indicating that ALM remains reliable even under challenging mask sizes.
>
> We would also like to emphasize again that our method is fully training-free. The **LoRA [C9]-based examples were included to demonstrate that ALM is flexible and can optionally benefit from fine-tuning techniques.** Here, upon the request of Reviewer’s suggestion, we additionally report the performance obtained by combining ALM with fine-tuned LoRA weight in Tables C10-C11. As expected, attaching LoRA yields a slight improvement (ALM w/ LoRA), while ALM already achieves strong performance without any training.
>
> Table C10. Quantitative results of ALM combined with LoRA weight on AFHQ dataset [C2]
>
> | Method      | LPIPS ↓ | MSE ↓  | SSIM ↑ | CS ↑  |
> |---------------|---------|--------|--------|-------|
> | ALM           | 0.391   | 0.142  | 0.591  | 29.00 |
> | ALM  w/ LoRA         | 0.384  | 0.128  |  0.590 |  29.12  |
>
> Table C11. Quantitative results of ALM combined with LoRA weight on CelebA-HQ dataset [C4]
>
> | Method      | LPIPS ↓ | MSE ↓  | SSIM ↑ | CS ↑  |
> |---------------|---------|--------|--------|-------|
> | ALM           | 0.342 | 0.125  | 0.644  | 28.01 |
> | ALM  w/ LoRA       |   0.318 |  0.111 | 0.654  | 27.05    |
>
>
>
> **Q5.** *Please report end-to-end runtime (with/without inversion).*
>
> **A10.** In the answer #3 of the global response, we analyze the computational cost of ALM and the baseline methods. Table C12 reports the required GPU memory (GB) and runtime (s) for generating a single image. Overall, **our method exhibits computational overhead comparable to existing baselines.** We note that although ALM incurs a slightly longer runtime than SyncSDE [C3] due to the additional sampling of $\epsilon_{\theta}(\mathbf{E}_t, t)$, the difference is minor and ALM achieves better performance than SyncSDE. With the results in Table 1 of the main paper, these findings highlight that **ALM achieves the best trade-off between performance and computational efficiency** among the various methods.
>
> Table C12. Computational cost analysis.
>
> | Method | Additional training cost  | [Sampling] GPU Memory (GB) | [Sampling] Runtime (s/image) | LPIPS (↓) |
> |--|--|--|--|--|
> | BrushNet  [G8]  | 3 days with 8 V100 GPUs  | 4.73   | 3.189  | 0.434 |
> | PowerPaint  [G9] | requires 8 A100 GPUs | 5.54  | 4.089  | 0.428 |
> | SyncSDE  [G6]  | - | 4.98   | 6.734   | 0.414 |
> | BLD   [G7]  | -  | 3.39 | 2.615    | 0.421 |
> | HD-Painter [G10]    | - | 29.07   | 38.840   | 0.398 |
> | ALM (Ours)    | - | 4.98 | 8.584   | 0.391 |
> | ALM w/o inversion| -   | 4.98   | 3.850   | - |
>
>
> **Q6.** *Please report a study of inversion error vs. observed-region PSNR/SSIM to demonstrate that preservation remains reliable under realistic noise/model choices.*
>
> **A11.** We measure SSIM on the observed regions and report the results for all methods in Table C13. As shown, our method achieves superior performance among training-free approaches and shows comparable results with training-based baselines. These results demonstrate that preservation of the observed content remains reliable.
>
> Regarding inversion error, we do not report it since our method doesn’t rely on reconstruction-based strategy. Nonetheless, we expect some discretization error in the diffusion models, and we expect that such errors can be mitigated using techniques like Null-text inversion [C18]. Exploring a tighter combination between ALM and exact reconstruction suggests a meaningful direction for future work.
>
> Table C13. Quantitative result of observed-SSIM on various datasets
>
> | Method      | Training-Free | AFHQ [C2] | CelebA-HQ [C4] | Seasons [C8] | Painters  [C8] |
> |----|---|--|----|---|--|
> | BrushNet   [C15]    |   N | 0.862 |  0.897   |   0.793  |   0.786  |
> | PowerPaint  [C16]    | N | 0.839   |   0.878  |   0.769  |   0.763  |
> | SyncSDE      [C3]   |  Y | 0.828 |   0.843  |  0.768  |   0.748  |
> | BLD        [C5]         |  Y | 0.844  |   0.885  |   0.780  |  0.765   |
> | HD-Painter    [C17]      |  Y | 0.824 |   0.857  |  0.769   |   0.760  |
> | ALM (Ours)         |  Y | 0.845 |  0.869   |    0.784  |   0.769  |

---

> ### Author Response · Authors · 2025-11-22
> **Response to Reviewer bmgX (5/5)**
>
> **Q7.** *For long-video generation, can you add FVD/Fréchet Video Distance and temporal consistency metrics, plus throughput (FPS) on standardized hardware, to substantiate the efficiency and coherence claims?*
>
> **A12.** We kindly request you to refer to A5 and Table C9, where we provide evaluations for long video generation. We use an FPS of 8 as the default setting for all video-related experiments.
>
>
> We hope our explanations have clarified the issues, and we welcome any further questions.
>
>
> **References**
>
> [C1] Kim et al., FIFO-Diffusion: Generating infinite videos from text without training. NeurIPS, 2024.
>
> [C2] Choi et al., StarGAN v2: Diverse image synthesis for multiple domains. CVPR, 2020.
>
> [C3] Lee et al., SyncSDE: A probabilistic framework for diffusion synchronization. CVPR, 2025.
>
> [C4] Karras et al., Progressive growing of GANs for improved quality, stability, and variation. ICLR, 2018.
>
> [C5] Avrahami et al., Blended latent diffusion. ACM TOG, 2023.
>
> [C6] Rombach et al., High-resolution image synthesis with latent diffusion models. CVPR, 2022.
>
> [C7] Podell et al., SDXL: Improving latent diffusion models for high-resolution image synthesis. ICLR, 2024.
>
> [C8] Anoosheh et al., ComboGAN: Unrestrained scalability for image domain translation. CVPRW, 2018.
>
> [C9] Hu et al., LoRA: Low-rank adaptation of large language models. ICLR, 2022.
>
> [C10] Qiu et al., FreeNoise: Tuning-free longer video diffusion via noise rescheduling. ICLR, 2024.
>
> [C11] Chen et al., SEINE: Short-to-long video diffusion model for generative transition and prediction. ICLR, 2023.
>
> [C12] Wang et al., LaVie: High-quality video generation with cascaded latent diffusion models. IJCV, 2025.
>
> [C13] Unterthiner et al., FVD: A new metric for video generation. ICLR Workshop, 2019.
>
> [C14] Radford et al., Learning transferable visual models from natural language supervision. ICML, 2021.
>
> [C15] Ju et al., BrushNet: A plug-and-play image inpainting model with decomposed dual-branch diffusion. ECCV, 2024.
>
> [C16] Zhuang et al., A task is worth one word: Learning with task prompts for high-quality versatile image inpainting. ECCV, 2024.
>
> [C17] Manukyan et al., HD-Painter: High-resolution and prompt-faithful text-guided image inpainting with diffusion models. ICLR, 2025.
>
> [C18] Mokady et al., Null-text inversion for editing real images using guided diffusion models. CVPR, 2023.

---

### Official Review · Reviewer_oARH · 2025-10-31

**Soundness:** 2
**Presentation:** 3
**Contribution:** 2
**Rating:** 4
**Confidence:** 3

**Summary:**

The authors propose a new diffusion synchronization algorithm that can be used to generate the missing regions in a given sample. They identify a weakness in previous diffusion synchronization methods, which only consider observed regions. Pairing the proposed algorithm with an acceleration strategy to reduce sampling time, the authors demonstrate notable improvements over previous methods on image inpainting/outpainting and human motion inpainting tasks.

**Strengths:**

- The authors address a clear limitation of previous diffusion synchronization approaches by adding a step to the sampling process that accounts for the unobserved regions. This is a novel contribution that aims to improve the quality of the generated images.

- In addition to proposing the "unobserved region step", the authors also propose an acceleration strategy to speed up sampling. They show that they can improve the unobserved region quality with a single-step approximation, instead of performing a large number of small, incremental steps.

- The method is showcased on very different modalities (images, human motion, video), which involves running it with multiple different diffusion models. By showing that it works across many tasks, the authors validate the proposed algorithm's wide applicability.

**Weaknesses:**

- The wide image generation results are difficult to interpret. The authors mention that they measure FID and KID, but they don't specify what the reference ground truth dataset is. Additionally, the quantitative and qualitative differences between the three methods are small, making it unclear whether there's a best method out of the three shown. I would also suggest measuring the FID on the patch level to show whether there are differences in the generated details.

- The long video generation experiment lacks any comparisons to baselines. The authors only provide qualitative results of their method on long video generation. Without establishing a baseline, it is difficult to understand what the argument is for long video generation. Does the proposed method perform better than naive outpainting?

- There are many mentions of the acceleration strategy, but it is never really tested in the experiments. The authors should have an ablation where they show how the original, slow, non-accelerated algorithm performs and compare it to their accelerated variant, which is the one they use in all experiments.

- There are no mentions of the sampling time in the experiments, which can play a critical role in training-free approaches. Table 1 should be complemented with a column outlining the compute requirements for each method (memory and runtime).

- I would suggest adding $\uparrow$ and $\downarrow$ symbols to the tables to improve readability. Since you have a lot of different experiments across many modalities, it will make it easier for a reader to follow your results.

**Questions:**

- Is there a point where the acceleration strategy breaks, e.g., if you use very large $w_1$ and $w_2$, does sampling diverge? Would the non-accelerated variant not break if you appropriately scaled it and performed the equivalent number of steps? Does FID improve with the non-accelerated variant and a large $N$?

- You mention that you apply DDIM inversion before running your algorithm to get the values of the $X_t$. Is the inversion applied to the masked image? How much additional overhead over the baselines that do not require inversion does this add?

- Should the mask in Figure 1 be flipped? In Section 3, the mask is used to denote the unobserved regions, while in Figure 1, it is the opposite.

---

> ### Author Response · Authors · 2025-11-22
> **Response to Reviewer oARH (1/2)**
>
> We thank you for spending valuable time reviewing our paper.
>
> **W1.** *The authors mention that they measure FID and KID, but they don't specify what the reference ground truth dataset is. Additionally, the quantitative and qualitative differences between the three methods are small. I would also suggest measuring the FID on the patch level to show whether there are differences in the generated details.*
>
> **A1.** For the reference image set used to compute FID and KID, we generate 2,000 images per text prompt using the pretrained Stable Diffusion model. As noted, we also measure FID and KID at the patch level to more precisely quantify how well each method enhances local details.
>
> For qualitative comparisons, we emphasize the key differences between our method and the baselines. In the left example (“Alpine village …”) of the Figure 6 of the main paper, SyncSDE [B1] exhibits noticeable color inconsistencies and edge artifacts across patch boundaries, whereas ALM clearly alleviates these issues. In the right example (“Nestled in a canyon …”), SyncTweedies [B2] produces blurry regions on the left side of the synthesized wide image, and SyncSDE shows structural inconsistency in the upper-right area. In contrast, ALM generates a wide image with neither blurred nor inconsistent regions.
>
>
> **W2.** *The long video generation experiment lacks any comparisons to baselines.*
>
> **A2.** In the answer #6 of the global response, we compare the proposed method for long video generation against three baselines; FreeNoise [B3], SEINE [B4], and SyncSDE [B1] using the pretrained LaVie model [B5]. For evaluation, we adopt FVD [B6], KVD, and CLIP [B7] text-video similarity. We use a total of 250 video sequences for evaluation, each containing 104 frames. We show the quantitative evaluation result in Table B1. As shown, our method outperforms the baselines, emphasizing the effectiveness of our method. Note that we scale the value of FVD and KVD by 1/1000. We also visualize the qualitative comparison in the Section G of the revised PDF file.
>
> Table B1. Quantitative evaluation of long video generation
>
> | Method      | FVD ↓    | KVD ↓    | CS ↑     |
> |-------------|----------|----------|----------|
> | FreeNoise [B3]  | 2.5516   | 4.3600   | 31.08   |
> | SEINE     [B4]  | 3.6496   | 6.6111   | 30.43   |
> | SyncSDE    [B1]  | 2.5049   | 5.6352   | **31.18**   |
> | ALM (Ours)  | **2.4870** | **4.2188** | 31.12 |
>
>
> **W3.** *There are many mentions of the acceleration strategy, but it is never really tested in the experiments.*
>
> **A3.** In the answer #4 of the global response, we analyze the benefit driven by the proposed acceleration strategy in terms of runtime. In Table B2, we show the effectiveness of the proposed acceleration strategy by measuring the runtime. **As shown, acceleration strategy leads to a significant reduction in runtime.** We additionally provide qualitative comparisons in the Section F of the revised PDF file, with and without the acceleration strategy. Notably, the visual quality remains consistent across both setups, demonstrating that the proposed acceleration strategy effectively reduces the computational cost while maintaining the overall performance.
>
> Table B2. Effectiveness of the proposed acceleration strategy in terms of computational cost
>
> | Method        | [Sampling] GPU Memory (GB) | [Sampling] Runtime (s/image) |
> |-------|----|----|
> | ALM (Ours)     | 4.98      | **8.58**       |
> | ALM w/o acceleration ($N=1000$)       | 4.98    | 1836.23     |
>
>
> **W4.** *There are no mentions of the sampling time in the experiments, which can play a critical role in training-free approaches.*
>
> **A4.** In the answer #3 of the global response, we analyze the computational cost of ALM and the baseline methods. Table B3 reports the required GPU memory (GB) and runtime (s) for generating a single image. Overall, **our method exhibits computational overhead comparable to existing baselines.** We note that although ALM incurs a slightly longer runtime than SyncSDE due to the additional sampling of $\epsilon_{\theta}(\mathbf{E}_t, t)$, the difference is minor and ALM achieves better performance than SyncSDE [B1]. With the results in Table 1 of the main paper, these findings highlight that **ALM achieves the best trade-off between performance and computational efficiency** among the various methods.
>
> Table B3. Computational cost analysis.
>
> | Method | Additional training cost  | [Sampling] GPU Memory (GB) | [Sampling] Runtime (s/image) | LPIPS (↓) |
> |--|--|--|--|--|
> | BrushNet  [B8]  | 3 days with 8 V100 GPUs  | 4.73 | 3.189 | 0.434 |
> | PowerPaint  [B9] | requires 8 A100 GPUs  | 5.54 | 4.089 | 0.428 |
> | SyncSDE  [B1]  | - | 4.98| 6.734 | 0.414 |
> | SyncSDE w/o inversion [B1]  | - | 4.98 | 6.734 | 0.672 |
> | BLD   [B10]  | -  | 3.39 | 2.615 | 0.421 |
> | HD-Painter [B11]  | - | 29.07 | 38.840   | 0.398 |
> | ALM (Ours) | - | 4.98 | 8.584 | **0.391** |
> | ALM w/o inversion  | - | 4.98  | 2.884  | 0.606  |

---

> ### Author Response · Authors · 2025-11-22
> **Response to Reviewer oARH (2/2)**
>
> **W5.** *I would suggest adding $\uparrow$ and $\downarrow$ symbols to the tables to improve readability.*
>
> **A5.** We thank you for the valuable feedback. We modified the manuscript accordingly and attached the revised PDF file.
>
> **Q1.** *Is there a point where the acceleration strategy breaks, e.g., if you use very large $w_1$ and $w_2$, does sampling diverge? Would the non-accelerated variant not break if you appropriately scaled it and performed the equivalent number of steps? Does FID improve with the non-accelerated variant and a large $N$?*
>
> **A6.** Yes, the sampling diverges when we use very large values of $w_1$ and $w_2$. As discussed in Section 3.4. of the main paper, our acceleration strategy is derived under the small-update assumption $|\Delta Y_t^i| << 1$ and the Lipschitz continuity [B12, B13] of the pretrained noise prediction network. These conditions ensure that the sequence $\{ \Delta Y_t^i \}_{i=1}^N$ remains approximately constant and can therefore be collapsed into a single step update (Eq. (24)). Consequently, when the assumption is violated by choosing excessively large $w_1$ and $w_2, the one-step approximation would no longer hold, and may produce unstable samples. We also note that the same issue would also arise for the non-accelerated variant if the corresponding hyperparameters were scaled to the same magnitude.
>
> In addition, we provide qualitative comparisons with and without the acceleration strategy in the Section F of the revised PDF. Notably, the visual quality remains consistent across both setups. Thus, we claim that metrics such as FID remain similar for the non-accelerated variant, while incurring significantly higher computational cost.
>
>
> **Q2.** *You mention that you apply DDIM inversion before running your algorithm to get the values of the $X_t$. Is the inversion applied to the masked image? How much additional overhead over the baselines that do not require inversion does this add?*
>
> **A7.** Yes, we apply the DDIM [B14] inversion to the masked source image for fair comparison. We also detail the computational cost of the baselines that do not require inversion in A4. We kindly request you to refer to Table B3.
>
>
> **Q3.** *Should the mask in Figure 1 be flipped? In Section 3, the mask is used to denote the unobserved regions, while in Figure 1, it is the opposite.*
>
> **A8.** Yes, the mask in Figure 1 should be flipped. We sincerely thank you for pointing this out. We modified the manuscript accordingly and attached the revised PDF file.
>
> We hope our explanations have clarified the issues, and we welcome any further questions.
>
>
> **References**
>
> [B1] Lee et al., SyncSDE: A probabilistic framework for diffusion synchronization. CVPR, 2025.
>
> [B2] Kim et al., SyncTweedies: A general generative framework based on synchronized diffusions. NeurIPS, 2024.
>
> [B3] Qiu et al., FreeNoise: Tuning-free longer video diffusion via noise rescheduling. ICLR, 2024.
>
> [B4] Chen et al., SEINE: Short-to-long video diffusion model for generative transition and prediction. ICLR, 2023.
>
> [B5] Wang et al., LaVie: High-quality video generation with cascaded latent diffusion models. IJCV, 2025.
>
> [B6] Unterthiner et al., FVD: A new metric for video generation. ICLR Workshop, 2019.
>
> [B7] Radford et al., Learning transferable visual models from natural language supervision. ICML, 2021.
>
> [B8] Ju et al., BrushNet: A plug-and-play image inpainting model with decomposed dual-branch diffusion. ECCV, 2024.
>
> [B9] Zhuang et al., A task is worth one word: Learning with task prompts for high-quality versatile image inpainting. ECCV, 2024.
>
> [B10] Avrahami et al., Blended latent diffusion. ACM TOG, 2023.
>
> [B11] Manukyan et al., HD-Painter: High-resolution and prompt-faithful text-guided image inpainting with diffusion models. ICLR, 2025.
>
> [B12] Karras et al., Elucidating the design space of diffusion-based generative models. NeurIPS, 2022.
>
> [B13] Kim et al., FIFO-Diffusion: Generating infinite videos from text without training. NeurIPS, 2024.
>
> [B14] Song et al., Denoising diffusion implicit models. ICLR, 2021.

---

> > ### Comment · Reviewer_oARH · 2025-11-26
> >
> > Thank you for your detailed responses. There are a few more questions I have after reading your rebuttal:
> >
> > > A1. Wide image FID/KID
> >
> > My question here was about the reference dataset with which you compute FID and KID. I understand that you generate images with your method, but you have to use some reference dataset statistics to compare with the generated image statistics for FID/KID. My issue is that for bigger images, since Inceptionv3 downsamples to 224x224, the FID does not capture local details. Can you clarify *how* you computed the FID? Which reference dataset did you use, and did you compute it on patches of the large images or the entire images?
> >
> > > A2. Video results
> >
> > In the new table B1 you provided, the FVD improvements look marginal, even when considering that you have scaled the FVD numbers by 1/1000. Are there noticeable differences between your generated videos and the baselines? It is quite hard to understand from Figure 17; in the smoke video, SEINE seems to obtain better consistency (the color of the smoke doesn't change), while in the bubbles video, your method seems to be the best.
> >
> > > A3. Runtime
> >
> > Thank you for adding this. I think that table B2 is not necessary for the main text -- it is quite clear that the accelerated version is faster as it runs with fewer steps. Figure 16 in the appendix provides a better overview of what the trade-off is between the accelerated and non-accelerated algorithms. Maybe consider moving that (or a smaller version of it) to the main text.
> >
> > > A4. Sampling time
> >
> > Thank you for adding this ablation. Is there a specific reason you did not include LPIPS without inversion? Does the algorithm not work at all if you do not run DDIM inversion?

---

> > > ### Author Response · Authors · 2025-11-27
> > >
> > > We sincerely thank you for additional questions regarding our initial response.
> > >
> > > **Q1.** *Wide image FID/KID*
> > >
> > > **A1.** We sincerely apologize for the lack of clarity in our previous explanation. We measure FID and KID on **on patches extracted from the generated wide images**, and the full evaluation protocol is detailed as follows.
> > >
> > > First, for each generated wide image of resolution $2048 \times 512$, we randomly crop a $512 \times 512$ patch by uniformly sampling the cropping location along the x-axis. Since our evaluation set contains 450 generated wide images, this process yields 450 cropped patches of size $512 \times 512$.
> > >
> > > Second, for each of the 9 prompts used in our evaluation, we generate 2,000 reference images using Stable Diffusion. For instance, for the prompt “A photo of a forest with a misty fog”, we synthesize 2,000 images at $512 \times 512$ resolution using the same prompt, and use them as the reference set for that category. This procedure is repeated for all nine prompts, producing nine distinct reference sets.
> > >
> > > Third, for each prompt, we compute FID/KID between the 50 cropped patches corresponding to that prompt and its associated 2,000 reference images. This results in nine FID/KID values, one for each prompt.
> > >
> > > Finally, we report the average of these nine values. The same evaluation procedure is applied consistently across all methods.
> > >
> > >
> > > **Q2.** *Video results*
> > >
> > > **A2.** Thank you for pointing this out. While SEINE [B4] maintains color consistency, we emphasize that ALM better preserves temporal structural consistency, as visualized in Figure 19 of the revised PDF. In the frame-by-frame visualization (Frames 10 - 15), ALM maintains a consistent structure that undergoes smooth and physically plausible deformation over time. The shape persists across consecutive frames, indicating strong temporal structural coherence. Conversely, SEINE exhibits abrupt structural reformation between adjacent frames, where distinct shapes appear and disappear without respecting the previous-frame geometry, thereby resulting in temporal discontinuities. Overall, ALM more faithfully preserves the temporal identity and evolution of the smoke structure.
> > >
> > > To show the differences between our method and baselines, we visualize additional qualitative results in Figure 20 of the revised PDF file.
> > > - For the case of the top example (“Wide shot of fire glowing …”), ALM demonstrates clear superiority in temporal coherence and visual stability. FreeNoise [B3] generates almost static sequences with minimal dynamics. SEINE [B4] fails to maintain color consistency, since its flame color abruptly shifts across frames and exhibits structural discontinuity. SyncSDE [B1] introduces noticeable artifacts (*e.g.*, at frame 80, top-left), which persist for 15 frames, indicating temporal artifact propagation. In contrast, **ALM preserves stable color, consistent geometry, and smooth temporal transition without artifacts, resulting in the most realistic long-video.**
> > >
> > > - In the bottom example (“Slow zoom onto a sparkler …”) ALM consistently produces the most stable and coherent video among all baselines. FreeNoise generates results that are generally comparable. SEINE again exhibits clear color inconsistency; that is, the flame and particle color shift significantly when comparing the first frame to later frames, indicating temporal drift. Moreover, SEINE introduces structural instability, where the sparkler head abruptly expands and collapses, leading to noticeable flickering. SyncSDE produces distinct artifacts, particularly evident in frames 40 and 80, where multiple blurry, blob-like particles appear. **ALM preserves consistent color temperature, maintains coherent particle trajectories, and avoids temporal artifacts.**

---

> ### Author Response · Authors · 2025-11-27
>
> **Q3.** *Runtime*
>
> **A3.** We thank the reviewer for the helpful suggestion. Following your recommendation, we have incorporated a smaller version of Figure 17 (originally, Figure 16) into the main text to clearly illustrate the trade-off between the accelerated and non-accelerated variants. We kindly request you to refer to Lines 357 - 370 in the revised PDF file. We believe this revision improves the clarity of the discussion.
>
>
> **Q4.** *Sampling time*
>
> **A4.** We did not report LPIPS values for the version of “without inversion” because that experiment was still underway, and we sincerely apologize for this. Our method can indeed operate without DDIM inversion by obtaining $\mathbf{X}_t$ through simple perturbation of $\mathbf{X}_0$. We updated Table 3 of the revised PDF and Table B3 of our response to include the LPIPS values for the inversion-free version. However, this approximation is less accurate than DDIM inversion, leading to degraded performance.
>
> The performance gets lower because directly perturbing $\mathbf{X}_0$ with noise provides only a single-step estimate of $\mathbf{X}_t$, whereas DDIM inversion reconstructs $\mathbf{X}_t$ **by tracing the exact deterministic trajectory of the pretrained diffusion model**. In diffusion models, each intermediate latent $\mathbf{X}_t$​ encodes not only noise but also the accumulated effect of multiple nonlinear denoising steps. Therefore, naively adding noise to obtain $\mathbf{X}_t$​ cannot replicate the model's true inversion dynamics and results in a latent representation that is misaligned with the reverse process. In contrast, DDIM inversion yields a more faithful and model-consistent estimate of $\mathbf{X}_t$, enabling more reliable guidance and substantially improving the performance. This phenomenon is well aligned with existing editing algorithms [B15, B16, B17], where DDIM inversion is widely adopted to obtain accurate intermediate latents.
>
> Both our method and SyncSDE rely on a guidance term during the reverse process to preserve the observed region, and the quality of this guidance is highly dependent on the accuracy of $\mathbf{X}_t$. The degradation observed in the “SyncSDE w/o inversion’’ row clearly demonstrates that this issue is not caused by the new components introduced in our method, but is inherent to the underlying SyncSDE mechanism itself. **Consequently, the use of DDIM inversion is not a limitation specific to our approach but a necessary requirement shared across existing diffusion-based editing methods.**
>
>
> **References**
>
> [B15] Parmar, et al. Zero-shot image-to-image translation. SIGGRAPH, 2023.
>
> [B16] Cao et al. Masactrl: Tuning-free mutual self-attention control for consistent image synthesis and editing. ICCV, 2023.
>
> [B17] Yang et al. FSI-Edit: Frequency and Stochasticity Injection for Flexible Diffusion-Based Image Editing. NeurIPS, 2025.

---

### Official Review · Reviewer_9Fyv · 2025-11-01

**Soundness:** 3
**Presentation:** 2
**Contribution:** 2
**Rating:** 4
**Confidence:** 2

**Summary:**

The paper proposes an update on the SyncSDE inpainting baseline by adding additional regularization to the sampling which maximizes the likelihood of the sampled region, in addition to the SyncSDE loss which forces the observed content to stay unchanged. This improves the performance of the training-free inpainting approach compared to SyncSDE. The evaluation on several datasets shows that this outperforms both trainng-free and training-based baselines on several datasets.

**Strengths:**

The paper identifies a weakness in the SyncSDE approach of training-free image inpainting by optimizing the trajectory such that the observed pixels stay unchanged. While this can lead to good results it does not explicitly enforce any consistency or realism on the inpainted pixels and instead just assumes that the inpainted pixels will look realistic due to the model's prior. This approach adds an additional optimization step that maximizes the conditional and unconditional likelihood of the inpainted pixels under the model's prior to make the inpainted pixels both agree with the observed pixels and make them look realistic overall. This is reasonable and seems to work.

The evaluation is done across several datasets and compares both training-free and training-based inpainting approaches. On the evaluated datasets the approach shows improved performance. The ablation study also shows the positive impact of each of the individual additions. It is especially nice that the evaluation was done across several domains, including images, videos, and 3D motion.

**Weaknesses:**

While the approach works it is a relatively small update/modification to an existing approach. The evaluation is also only done on relatively simple datasets (e.g., CelebA-HQ and AFHQ) which might be too simple for today's SOTA models and even out of domain for some of the baselines. The video examples also only show very simplistic video motions.
I also did not see any comparison around the increased cost of this approach, which to my understanding, requires additional model evaluations for each sampling step, as well as DDIM inversion of the input.

**Questions:**

There seem to be some additional weighting hyperparameters introduced by this approach. How do they affect the outcome and how susceptible is the approach to the values of those hyperparameters?
Which baseline model was used for ALM?
How well does this approach work on more complicated images and how well does it work on larger baseline models for image and video generation?
What is the sampling speed and incurred additional sampling cost of this approach compared to the other training-free baselines?

---

> ### Author Response · Authors · 2025-11-22
> **Response to Reviewer 9Fyv (1/2)**
>
> We thank you for spending valuable time reviewing our paper.
>
> **W1.** *While the approach works it is a relatively small update/modification to an existing approach.*
>
> **A1.** In the answer #1 of the global response, we recap the novelty and significance of the proposed method. **We emphasize that ALM provides a fundamentally different and highly practical advantage over existing approaches: it enables high-fidelity versatile content generation entirely without training, operating directly on top of any pretrained foundation model.** Unlike prior methods that require heavy task-specific training and large datasets, ALM instantly transforms generative models such as SD [A1] or SDXL [A2] into powerful versatile content generation models with zero additional computational cost.
>
>
> **Q2**. *How well does this approach work on more complicated images and how well does it work on larger baseline models for image and video generation?*
>
> **A2.** In the answer #5 of the global response, we discuss the application of ALM beyond simple scenarios (including image and video). Specifically, we show the case of complicated images with complex text prompts and videos with dynamic motions. We also show the qualitative results in the Section E of the revised PDF.
>
> In addition, in the answer #1 of the global response, we show the performance of ALM using the relatively large image generation model (SDXL [A2]), which consistently works well regardless of the baseline model. To show that ALM performs well on the large baseline model such as SDXL, we combine ALM to SDXL and quantitatively compare against SDXL-Inpainting (Table A1-A2) using the same setup described in the main paper. Our method achieves comparable performance compared to SDXL-Inpainting. **This demonstrates that ALM can immediately upgrade pretrained image generation models to perform high-fidelity versatile content generation purely through our training-free mechanism, even on larger baseline models.**
>
> Table A1. Quantitative comparison of the proposed method with SDXL-Inpainting on CelebA-HQ dataset [A3].
>
> | Method      | Training-free | LPIPS ↓ | MSE ↓  | SSIM ↑ | CS ↑   |
> |-------|-----|------|-----|-----|-----|
> | SDXL-Inpainting  | N  | **0.286**  | **0.046**  | 0.693  | 27.11 |
> | ALM (Ours)       | Y     | 0.315   | 0.105  | **0.710**  | **27.78** |
>
> Table A2. Quantitative comparison of the proposed method with SDXL-Inpainting on Painters dataset [A4].
>
> | Method    | Training-free | LPIPS ↓ | MSE ↓  | SSIM ↑ | CS ↑   |
> |-------|-----|----|------|------|----|
> | SDXL-Inpainting  | N      | **0.338**   | **0.052**  | 0.609  | 26.77 |
> | ALM (Ours)       | Y   | 0.422   | 0.133  | **0.618** |**27.50**
>
>
> **Q3.** *There seem to be some additional weighting hyperparameters introduced by this approach. How do they affect the outcome and how susceptible is the approach to the values of those hyperparameters?*
>
> **A3**. In the answer #2 of the global response, we analyze the effects of each hyperparameter and show that our method is robust to the change of hyperparameters. Specifically, we sweep $w_1$ over $[0.5, 1, 1.5]$, and $w_2$ over $[0.001, 0.005, 0.01]$ and provide the corresponding quantitative results in Table A3-A4 as well as qualitative comparisons in the Section D of the revised PDF. Note that the default setting used in the main paper is $w_1 = 1$ and $w_2 = 0.005$. As shown, our method consistently maintains strong performance across all tested configurations, still outperforming the baseline methods, thereby confirming the robustness of the proposed method.
>
> Table A3. Quantitative analysis of the hyperparameter sensitivity on AFHQ dataset [A5].
>
> | Method                          | LPIPS ↓ | MSE ↓  | SSIM ↑ | CS ↑  |
> |------|-----|----|----|----|
> | Baseline (SyncSDE [A6], Best)     |  0.414   |  0.172  | 0.552  |  28.80 |
> | ALM ($w_1=1.0$, $w_2=0.001$)     | **0.388** | 0.146  | 0.582  | 28.91 |
> | ALM ($w_1=1.0$, $w_2=0.005$)     | 0.391   | 0.142  | 0.591  | 29.00 |
> | ALM ($w_1=1.0$, $w_2=0.01$ )   | 0.400   | 0.138  | **0.597** | **29.07** |
> | ALM ($w_1=0.5$, $w_2=0.005$)     | 0.402   | 0.159  | 0.578  | 28.82 |
> | ALM ($w_1=1.5$, $w_2=0.005$ )     | 0.389   | **0.136** | 0.594  | 29.04 |
>
>
>
> Table A4. Quantitative analysis of the hyperparameter sensitivity on CelebA-HQ dataset [A3].
>
> | Method                          | LPIPS ↓ | MSE ↓  | SSIM ↑ | CS ↑  |
> |---|-----|----|----|-----|
> | Baseline (BLD [A7], Best)       | 0.372   | 0.130  | 0.611  | **28.64** |
> | ALM ($w_1=1.0$, $w_2=0.001$ )    | **0.340** | 0.129  | 0.634  | 28.04 |
> | ALM ($w_1=1.0$, $w_2=0.005$  )  | 0.342   | 0.125  | 0.644  | 28.01 |
> | ALM ($w_1=1.0$, $w_2=0.01$  )      | 0.349   | 0.123  | **0.651** | 27.99 |
> | ALM ($w_1=0.5$, $w_2=0.005$ )      | 0.359   | 0.137  | 0.627  | 28.05 |
> | ALM ($w_1=1.5$, $w_2=0.005$ )     | **0.340** | **0.121** | 0.648  | 28.04 |

---

> ### Author Response · Authors · 2025-11-22
> **Response to Reviewer 9Fyv (2/2)**
>
> **Q4.** *I also did not see any comparison around the increased cost of this approach, which to my understanding, requires additional model evaluations for each sampling step, as well as DDIM inversion of the input.. What is the sampling speed and incurred additional sampling cost of this approach compared to the other training-free baselines?*
>
> **A4.** In the answer #3 of the global response, we analyze the computational cost of ALM and the baseline methods. Table A5 reports the required GPU memory (GB) and runtime (s) for generating a single image. Overall, **our method exhibits computational overhead comparable to existing baselines.** We note that although ALM incurs a slightly longer runtime than SyncSDE [A6] due to the additional sampling of $\epsilon_{\theta}(\mathbf{E}_t, t)$, the difference is minor and ALM achieves better performance than SyncSDE. With the results in Table 1 of the main paper, these findings highlight that **ALM achieves the best trade-off between performance and computational efficiency** among the various methods.
>
> Table A5. Computational cost analysis.
>
> | Method | Additional training cost  | [Sampling] GPU Memory (GB) | [Sampling] Runtime (s/image) | LPIPS (↓) |
> |--|--|--|--|--|
> | BrushNet  [G8]  | 3 days with 8 V100 GPUs  | 4.73   | 3.189  | 0.434 |
> | PowerPaint  [G9] | requires 8 A100 GPUs | 5.54  | 4.089  | 0.428 |
> | SyncSDE  [G6]  | - | 4.98   | 6.734   | 0.414 |
> | BLD   [G7]  | -  | 3.39 | 2.615    | 0.421 |
> | HD-Painter [G10]    | - | 29.07   | 38.840   | 0.398 |
> | ALM (Ours)    | - | 4.98 | 8.584   | 0.391 |
> | ALM w/o inversion| -   | 4.98   | 3.850   | - |
>
>
> **Q5.** *Which baseline model was used for ALM?*
>
> **A5.** For image inpainting, we use 1) the pretrained Unconditional diffusion model [A11] trained on CelebA-HQ dataset [A3], 2) the pretrained Stable Diffusion v1-5 [A1], and 3) the pretrained Stable Diffusion XL [A2]. In the case of wide image generation, we use the pretrained Stable Diffusion v2-1-base model. For human motion inpainting, we use the pretrained U-Net-based human motion diffusion model discussed in *Karunratanakul et al* [A12]. Lastly, we use the pretrained LaVie model [A13] for long video generation.
>
>
> We hope our explanations have clarified the issues, and we welcome any further questions.
>
> **References**
>
> [A1] Rombach et al., High-resolution image synthesis with latent diffusion models. CVPR, 2022.
>
> [A2] Podell et al., SDXL: Improving latent diffusion models for high-resolution image synthesis. ICLR, 2024.
>
> [A3] Karras et al., Progressive growing of GANs for improved quality, stability, and variation. ICLR, 2018.
>
> [A4] Anoosheh et al., ComboGAN: Unrestrained scalability for image domain translation. CVPRW, 2018.
>
> [A5] Choi et al., StarGAN v2: Diverse image synthesis for multiple domains. CVPR, 2020.
>
> [A6] Lee et al., SyncSDE: A probabilistic framework for diffusion synchronization. CVPR, 2025.
>
> [A7] Avrahami et al., Blended latent diffusion. ACM TOG, 2023.
>
> [A8] Ju et al., BrushNet: A plug-and-play image inpainting model with decomposed dual-branch diffusion. ECCV, 2024.
>
> [A9] Zhuang et al., A task is worth one word: Learning with task prompts for high-quality versatile image inpainting. ECCV, 2024.
>
> [A10] Manukyan et al., HD-Painter: High-resolution and prompt-faithful text-guided image inpainting with diffusion models. ICLR, 2025.
>
> [A11] Lugmayr et al., RePaint: Inpainting using denoising diffusion probabilistic models. CVPR, 2022.
>
> [A12] Karunratanakul et al., Guided motion diffusion for controllable human motion synthesis. ICCV, 2023.
>
> [A13] Wang et al., LaVie: High-quality video generation with cascaded latent diffusion models. IJCV, 2025.

---

### Author Response · Authors · 2025-11-22
**Global Response (4/4)**

**References**

[G1] Rombach et al. High-resolution image synthesis with latent diffusion models. CVPR, 2022.

[G2] Podell et al. SDXL: Improving latent diffusion models for high-resolution image synthesis. ICLR, 2024.

[G3] Karras et al. Progressive growing of GANs for improved quality, stability, and variation. ICLR, 2018.

[G4] Anoosheh et al. ComboGAN: Unrestrained scalability for image domain translation. CVPRW, 2018.

[G5] Choi et al. StarGAN v2: Diverse image synthesis for multiple domains. CVPR, 2020.

[G6] Lee et al. SyncSDE: A probabilistic framework for diffusion synchronization. CVPR, 2025.

[G7] Avrahami et al. Blended latent diffusion. ACM TOG, 2023.

[G8] Ju et al. BrushNet: A plug-and-play image inpainting model with decomposed dual-branch diffusion. ECCV, 2024.

[G9] Zhuang et al. A task is worth one word: Learning with task prompts for high-quality versatile image inpainting. ECCV, 2024.

[G10] Manukyan et al. HD-Painter: High-resolution and prompt-faithful text-guided image inpainting with diffusion models. ICLR, 2025.

[G11] Qiu et al. FreeNoise: Tuning-free longer video diffusion via noise rescheduling. ICLR, 2024.

[G12] Chen et al. SEINE: Short-to-long video diffusion model for generative transition and prediction. ICLR, 2023.

[G13] Wang et al. LaVie: High-quality video generation with cascaded latent diffusion models. IJCV, 2025.

[G14] Unterthiner, Thomas, et al. FVD: A new metric for video generation. ICLR Workshop, 2019.

[G15] Radford et al. Learning transferable visual models from natural language supervision. ICML, 2021.

---

### Author Response · Authors · 2025-11-22
**Global Response (3/4)**

**3. Analysis on computational cost (R1, R2, R3, R4)**

We quantitatively evaluate the computational cost of ALM in comparison to baseline methods. Table G7 reports the required GPU memory (GB) and runtime (s) for generating a single image. We additionally include the runtime of ALM excluding the preprocessing stage (DDIM inversion). Overall, **our method exhibits computational overhead comparable to existing baselines.** We note that although ALM incurs a slightly longer runtime than SyncSDE [G6] due to the additional sampling of $\epsilon_{\theta}(\mathbf{E}_t, t)$, the difference is minor and ALM achieves better performance than SyncSDE. With the results in Table 1 of the main paper, these findings highlight that **ALM achieves the best trade-off between performance and computational efficiency** among the various methods.

Table G7. Computational cost analysis.

| Method | Additional training cost  | [Sampling] GPU Memory (GB) | [Sampling] Runtime (s/image) | LPIPS (↓) |
|--|--|--|--|--|
| BrushNet  [G8]  | 3 days with 8 V100 GPUs  | 4.73   | 3.189  | 0.434 |
| PowerPaint  [G9] | requires 8 A100 GPUs | 5.54  | 4.089  | 0.428 |
| SyncSDE  [G6]  | - | 4.98   | 6.734   | 0.414 |
| BLD   [G7]  | -  | 3.39 | 2.615    | 0.421 |
| HD-Painter [G10]    | - | 29.07   | 38.840   | 0.398 |
| ALM (Ours)    | - | 4.98 | 8.584   | 0.391 |
| ALM w/o inversion| -   | 4.98   | 3.850   | - |



**4. Ablation study on the proposed acceleration strategy (R2)**

We thank R2 for highlighting the important ablation study. We firstly clarify that the terminology *acceleration* refers to the one-step approximation introduced in Eq. (9) of the main paper, rather than a reduction in the overall sampling steps of the pretrained diffusion model.

In Table G8, we show the effectiveness of the proposed acceleration strategy by measuring the runtime. **As shown, acceleration strategy leads to a significant reduction in runtime.** We additionally provide qualitative comparisons in the Section F of the revised PDF file, with and without the acceleration strategy. Notably, the visual quality remains consistent across both setups, demonstrating that the proposed acceleration strategy effectively reduces the computational cost while maintaining the overall performance.

Table G8. Effectiveness of the proposed acceleration strategy in terms of computational cost

| Method  | [Sampling] GPU Memory (GB) | [Sampling] Runtime (s/image) |
|-|----|--|
| ALM (Ours)  | 4.98 | **8.58**  |
| ALM w/o acceleration ($N=1000$) | 4.98   | 1836.23    |


**5. Application of ALM on complex scenarios (R1)**

We address the point raised by R1 by demonstrating that ALM is readily applicable to both images and videos that contain rich, complex information. In the Section E of the revised PDF, we qualitatively report ALM’s performance on challenging scenarios. We first provide visualizations of long video generation results with dynamic motions, further highlighting the versatility of our method. We then show that our method remains high performance even when the source prompt is extremely long and packed with semantic details.


**6. Quantitative evaluation on long video generation (R2, R3)**

We compare the proposed method for long video generation against three baselines; FreeNoise [G11], SEINE [G12], and SyncSDE [G8] using the pretrained LaVie model [G13]. For evaluation, we adopt FVD [G14], KVD, and CLIP [G15] text-video similarity. We use a total of 250 video sequences for evaluation, each containing 104 frames. We show the quantitative evaluation result in Table G9. As shown, our method outperforms the baselines, emphasizing the effectiveness of our method. Note that we scale the value of FVD and KVD by 1/1000. We also visualize the qualitative comparison in the Section G of the revised PDF file.

Table G9. Quantitative evaluation of long video generation

| Method      | FVD ↓    | KVD ↓    | CS ↑     |
|--|---|---|---|
| FreeNoise [G11]  | 2.5516   | 4.3600   | 31.08   |
| SEINE   [G12]    | 3.6496   | 6.6111   | 30.43   |
| SyncSDE   [G8]  | 2.5049   | 5.6352   | **31.18**   |
| ALM (Ours)  | **2.4870** | **4.2188** | 31.12 |

---

### Author Response · Authors · 2025-11-22
**Global Response (2/4)**

## Response to reviewers

We now clarify the common points carried out by reviewers in detail.

**1. Significance of ALM (R1, R4)**

Recent visual foundation models provide versatile content generation capabilities, including inpainting and outpainting, as pointed out by R4. However, they typically require substantial task-specific and computational intensive training on large datasets, which is rarely open-sourced, limiting reproducibility and practical deployment.

In contrast, **ALM is fully training-free and operates entirely on top of any pretrained foundation model, effectively extending the applicability of modern foundation models beyond generation.** For example, ALM transforms powerful generative models (*e.g.*, SD [G1], SDXL [G2]) into highly capable content-editing models without any additional training. This completely eliminates the heavy computational required by existing approaches.

We apply ALM to the widely-used diffusion models and compare against SD-Inpainting (Table G1-G2), and SDXL-Inpainting (Table G3-G4). Our method outperforms SD-Inpainting, and achieves comparable performance compared to SDXL-Inpainting. **This demonstrates that ALM can immediately upgrade pretrained image generation models to perform high-fidelity versatile content generation purely through our training-free mechanism.**

Table G1. Quantitative comparison of the proposed method with SD-Inpainting on CelebA-HQ dataset [G3].

| Method  | Training-free | LPIPS ↓ | MSE ↓ | SSIM ↑ | CS ↑ |
|-|-|-|-|-|-|
| SD-Inpainting | N   | 0.356   | 0.130  | 0.570  | 26.29  |
| ALM (Ours) | Y | **0.342** | **0.125**  | **0.644** | **28.01**  |

Table G2. Quantitative comparison of the proposed method with SD-Inpainting on Painters dataset [G4].

| Method  | Training-free | LPIPS ↓ | MSE ↓ | SSIM ↑ | CS ↑   |
|-|-|-|-|-|-|
| SD-Inpainting | N   | **0.452**   | 0.163  | 0.391  | 26.06 |
| ALM (Ours) | Y  | 0.504 | **0.153** |**0.442**  | **27.26** |

Table G3. Quantitative comparison of the proposed method with SDXL-Inpainting on CelebA-HQ dataset.

| Method | Training-free | LPIPS ↓ | MSE ↓  | SSIM ↑ | CS ↑   |
|-|-|-|-|-|-|
| SDXL-Inpainting  | N  | **0.286**  | **0.046**  | 0.693 | 27.11 |
| ALM (Ours) | Y | 0.315 | 0.105 | **0.710** |**27.78**|

Table G4. Quantitative comparison of the proposed method with SDXL-Inpainting on Painters dataset.

| Method| Training-free | LPIPS ↓ | MSE ↓  | SSIM ↑ | CS ↑   |
|-|-|-|-|-|-|
| SDXL-Inpainting  | N | **0.338** |**0.052**  | 0.609  | 26.77 |
| ALM (Ours) | Y | 0.422   | 0.133  | **0.618** |**27.50** |

Also ALM operates entirely on open-source models and does not rely on closed training pipelines, every step of the method is reproducible. To emphasize our contribution, **we plan to release the official implementation of ALM upon the acceptance**, ensuring accessibility for the research community.

In summary, ALM provides a **training-free, open, and widely applicable mechanism** that turns any pretrained diffusion based foundation generative model into a flexible content editing model, without extra compute, and with full reproducibility.


**2. Analysis on hyperparameter sensitivity (R1, R3, R4)**

In the main paper, we introduce three hyperparameters: $w_1$, $w_2$ and $w_3$. In practice, we fix $w_1 = w_3$, leaving two tunable parameters $w_1$ and $w_2$.  **We now demonstrate that ALM is robust under variations of these hyperparameters**. We sweep $w_1 \in [0.5, 1, 1.5]$ and $w_2 \in [0.001, 0.005, 0.01]$ and provide the corresponding quantitative results in Table G5-G6 as well as qualitative comparisons in the Section D of the revised PDF. Note that the default setting used in the main paper is $w_1 = 1$ and $w_2 = 0.005$. As shown, our method consistently maintains strong performance across all tested configurations.

Table G5. Quantitative analysis of the hyperparameter sensitivity on AFHQ dataset [G5].

| Method | LPIPS ↓ | MSE ↓  | SSIM ↑ | CS ↑  |
|-|---|--|--|-|
| Baseline (SyncSDE [G6], Best)|  0.414   |  0.172  | 0.552  |  28.80 |
| ALM ($w_1=1.0$, $w_2=0.001$)  | **0.388** | 0.146  | 0.582  | 28.91 |
| ALM ($w_1=1.0$, $w_2=0.005$) | 0.391   | 0.142  | 0.591  | 29.00 |
| ALM ($w_1=1.0$, $w_2=0.01$ ) | 0.400   | 0.138  | **0.597** | **29.07** |
| ALM ($w_1=0.5$, $w_2=0.005$) | 0.402   | 0.159  | 0.578  | 28.82 |
| ALM ($w_1=1.5$, $w_2=0.005$ )  | 0.389   | **0.136** | 0.594  | 29.04 |


Table G6. Quantitative analysis of the hyperparameter sensitivity on CelebA-HQ dataset [G3].

| Method | LPIPS ↓ | MSE ↓  | SSIM ↑ | CS ↑ |
|-|-|-|-|-|
| Baseline (BLD [G7], Best)   | 0.372   | 0.130  | 0.611  | **28.64** |
| ALM ($w_1=1.0$, $w_2=0.001$ ) | **0.340** | 0.129  | 0.634  | 28.04 |
| ALM ($w_1=1.0$, $w_2=0.005$  ) | 0.342   | 0.125  | 0.644  | 28.01 |
| ALM ($w_1=1.0$, $w_2=0.01$  )  | 0.349   | 0.123  | **0.651** | 27.99 |
| ALM ($w_1=0.5$, $w_2=0.005$ ) | 0.359   | 0.137  | 0.627  | 28.05 |
| ALM ($w_1=1.5$, $w_2=0.005$ )| **0.340** | **0.121** | 0.648  | 28.04 |

---

### Author Response · Authors · 2025-11-22
**Global Response (1/4)**

[For convenience, we denote each reviewer as: **R1: 9Fyv, R2: oARH, R3: bmgX, R4: bVRd.**]

We thank you for spending valuable time reviewing our paper. Our paper effectively addresses the limitations of previous diffusion synchronization methods by adding an optimization procedure to the sampling process of the unobserved regions, which is configured by reviewers as “reasonable (R1)”, “novel (R2)”, “well-motivated (R3)”, and “highly robust and generalizable (R4)”. Our work is training-free, and widely applicable to diverse modalities including image, video, and human domain, which is demonstrated with the “nice (R1)” and “compelling (R4)” experiments.


## Manuscript Modification

*Main paper*
- Introduction: We revised the Introduction to more clearly highlight the key contributions and significance of our work.
- Method: We corrected the mask visualization in Figure 1.
- Experiments
   - We revised the table to incorporate $\uparrow$ and $\downarrow$ indicators for improved readability.
   - We added an analysis of computational cost.
   - We revised the caption of Figure 5 to better describe the qualitative results for wide image generation.
- Overall: We fixed several citation issues.

*Appendix*
- Section A: We present the quantitative comparison between our method and SDXL-Inpainting (Global Response #1).
- Section D: We visualize results from our hyperparameter sensitivity analysis (Global Response #2).
- Section E: We present additional image inpainting and long-video generation examples under challenging conditions, including extremely long source prompts and videos with significant dynamic motion (Global Response #5).
- Section F: We include analysis demonstrating the effectiveness of the proposed acceleration strategy (Global Response #4).
- Section G: We provide comparisons between ALM and baseline methods on long video generation tasks (Global Response #6).

## Summary of Global Response

We highlight that ALM is a **fully training-free method** that extends the applicability of pretrained diffusion based foundation models (e.g., SD, SDXL) to high-fidelity content editing tasks, including inpainting and outpainting. Below, we summarize our global response:
- **Zero Training Overhead**: ALM requires no task-specific training, large datasets, or proprietary tooling, yet achieves performance comparable to or exceeding existing training-based methods (Global Response 1). This makes it immediately deployable and fully reproducible.
- **Robustness and Efficiency**: ALM maintains strong performance across a wide range of hyperparameters (Global Response #2) and achieves a favorable trade-off between quality and computational cost (Global Response #3). Our acceleration strategy reduces runtime by orders of magnitude while preserving visual fidelity (Global Response #4).
- **Versatility Across Complex Scenarios**: ALM successfully handles images and long videos with rich semantic details and dynamic motion. Our method reliably supports prompt-based editing with extended, complex textual descriptions and long video with dynamic motion (Global Response #5). Additionally, quantitative evaluation on 250 long video sequences shows superior FVD/KVD performance over strong baselines (Global Response #6).

---

### Author Response · Authors · 2025-11-24
**Manuscript revised**

Dear Reviewers,

Thank you very much for your constructive feedback on our work. We have carefully revised the manuscript to address all of your comments and suggestions. Please feel free to take a look at the updated version along with our detailed responses.

Sincerely,

Authors

---

### Author Response · Authors · 2025-12-02
**Final Summary of Author Responses**

We thank the discussion and feedback regarding our submission. We provide a clear summary of our responses and revisions addressing the major concerns raised by the Reviewers.

**1. Significance of work and applicability to complex scenarios**

- We highlight that ALM is significant because it extends the capabilities of recent foundation models to a wide range of versatile content generation tasks without requiring additional training. Our experimental results demonstrate that ALM effectively handles versatile image generation with complex text and long video synthesis tasks.

**2. Robustness on Hyperparameter values**

- We conducted experiments with various hyperparameter configurations and showed that our method remains robust, consistently outperforming baselines regardless of hyperparameter changes.

**3. Computational cost analysis**

- We analyzed the computation cost of ALM and baselines in terms of required memory and runtime. The results show that ALM achieves competitive trade-off between performance and computational cost

**4. Effectiveness of acceleration strategy**


- We clarified that the proposed acceleration strategy significantly reduces the computation requirements while maintaining the overall performance.

**5. Evaluation of long video generation**

- We showed that ALM outperforms state-of-the-art baselines, both quantitatively and qualitatively, on long video generation across 250 diverse video sequences.

We have made every effort to address all concerns raised by the reviewers, and the corresponding revisions are reflected in both our responses and the revised manuscript. We thank you once again for SAC, AC, and Reviewer’s time and valuable feedback.

---

### Meta-Review · Area_Chair_dnRg · 2025-12-19

**Summary:**

The paper introduces Accelerated Likelihood Maximization (ALM), a training-free sampling strategy designed to improve consistency in versatile content generation by optimizing unobserved regions. While reviewers acknowledged the method's potential, the submission faced several criticisms for significant evaluation deficiencies, including missing computational cost analyses and quantitative baselines for video generation. Furthermore, there is criticism that the method is incremental; one reviewer noted it represents a relatively small update/modification to the existing SyncSDE framework rather than a significant methodological leap. Three out of four reviewers gave a score of borderline rejection, finding the initial submission incomplete and the contribution limited. Although the authors provided a substantial amount of new experimental data during the rebuttal, the need of adding such fundamental evaluations at this stage implies that the original submission was premature. Given the combination of incremental novelty and the need for a full review cycle to validate the extensive new results, the paper is not currently ready for publication.

**Reviewer Concerns:**

Concerns that were addressed in the rebuttal :
- The authors made a substantial effort to generate missing data including computational cost analysis.
- Ablation studies on weights ($w_1, w_2$) were provided.
- Comparisons against FreeNoise and SEINE (FVD/KVD metrics) were introduced for the long-video tasks.
- New results comparing ALM to SDXL-inpainting were added.

Concerns not addressed or partially addressed:
- Despite the volume of new results, several core concerns remain or were exacerbated by the rebuttal. Even with new baselines, improvements in video metrics (FVD) appeared marginal and raised valid methodological questions regarding how FID/KVD was calculated on patches for wide images.
- Concerns about the reliance on strong approximations (small-update assumption) and heuristic factorizations without verifiable conditions remain a weakness in the method's formulation.

The primary outstanding concern is the sheer volume of fundamental evaluation data (runtime, basic baselines, sensitivity analysis) introduced only during the rebuttal. This indicates the initial submission was premature. Properly validating these extensive new results requires a fresh review cycle rather than a quick check during the discussion

**Reviewer Scores:**

- Reviewers 9Fyv, oARH, and bmgX. These reviewers scored the paper as a borderline reject. They identified the paper as incomplete. While the authors attempted to fill the gaps, the marginal nature of the new improvements and the late arrival of fundamental data would probably still support a rejection from these reviewers.

- Reviewer bVRd: May maintain a weak accept, though their concern regarding significance against SOTA foundation models remains relevant.

---

### Decision · Program_Chairs · 2026-01-26

Reject